# Flexible changes to the Heliothis virescens ascovirus 3h (HvAV-3h) virion components affect pathogenicity against different host larvae species

Huan Yu,[1] Hong Chen,[1] Ni Li,[1] Chang-Jin Yang,[1,2] Hua-Yan Xiao,[1] Gong Chen,[1] Guo-Hua Huang[1]

**ABSTRACT**   The pathogenicity of a virus to a specific host species is an inerratic and describable ability of a virus to cause infection but is generally shaped by a variety of abiotic and biotic factors. In this investigation, the variations in pathogenicity of Heliothis virescens ascovirus 3h (HvAV-3h) to five noctuid pests were assessed based on mass spectrometry analysis on the virion compositions. Twenty-nine common HvAV-3h proteins were shared across all hosts, and different flexible proteins were identified in the virions of each specific host. Different host proteins were identified as HvAV-3h virion-associated proteins, including different detoxification enzyme proteins. Furthermore, a relatively fixed relationship between viral replication and changes in host detoxification enzyme activity caused by deficiencies in various viral structural proteins was found in the host larvae using a correlation matrix analysis: the host larval carboxylesterase and cytochrome P450 monooxygenases generally had highly similar responses to the viruses blocked by different structural proteins' antisera and their effects on viral DNA replication. Different interaction patterns for the virion structural proteins were found in different host larvae-produced virions, and the interactions between *Spodoptera litura* glutathione S-transferases and viral structural proteins were confirmed. The different host responses after viral infection could be the reason for the changes in viral pathogenicity, while the virus responses gradually adapted to the different hosts and there were flexible changes in the virion structures.

**IMPORTANCE**   Different pathogenic processes of a virus in different hosts are related to the host individual differences, which makes the virus undergoes different survival pressures. Here, we found that the virions of an insect virus, Heliothis virescens ascovirus 3h (HvAV-3h), had different protein composition when they were purified from different host larval species. These "adaptive changes" of the virions were analyzed in detail in this study, which mainly included the differences of the protein composition of virions and the differences in affinity between virions and different host proteins. The results of this study revealed the flexible changes of viruses to help themselves adapt to different hosts. Also, these interesting findings can provide new insights to improve our understanding of virus adaptability and virulence differentiation caused by the adaptation process.

**KEYWORDS**   insect virus, ascovirus, Heliothis virescens ascovirus 3h (HvAV-3h), pathogenicity, structural protein

Compared to bacteria and fungi, the structures of viruses are relatively simple, and they are also more adaptable and experience faster rates of mutation (1, 2). To produce a large number of progeny in a short time, viruses must complete genomic synthesis and virion assembly as rapidly as possible. Viral replication processes are

Address correspondence to Gong Chen, gongchen105@163.com, or Guo-Hua Huang, ghhuang@hunau.edu.cn.

The authors declare no conflict of interest.

See the funding table on p. 22.

often accompanied by complex host responses, including numerous gene and material exchanges between the hosts and viruses (3, 4). Viruses must thus also overcome the selection pressures in their various host environments, and to achieve this, they need to carry out the required self-improvements to facilitate rapid adaptations to their "new environment" (5).

Ascoviruses are insect viruses with double-stranded genomic DNA belonging to the *Ascoviridae* family, which was established in 2000, making it one of the newest virus families, and they reportedly attack the larvae of lepidopterous insects (6). In the field, ascoviruses are transmitted by the oviposition behavior of parasitic wasps, as once the parasitic wasps lay eggs in ascovirus-infected larvae, they become ascovirus carriers that can then inoculate other healthy larvae via their contaminated ovipositors (7, 8). A typical symptom characterizing ascovirus infection in the host larvae is the change in color of the hemolymph from clear and transparent to milky white, which is a result of the accumulation of numerous vesicles with rod-shaped virions in the host larval hemolymph (9, 10).

Compared to baculoviruses, which are well-known class of insect virus, ascoviruses have a relatively broad host spectra, as baculoviruses usually only infect one or several closely related insect larval species (11–14), whereas ascoviruses, especially the species *Heliothis virescens ascovirus* (HvAV), can infect several families of insect larvae (15–17). One ascovirus isolate, in particular, showed significant differences in its pathogenesis of different host larvae during long-term laboratory tests (17, 18), which highlights that the prolonged survival times for ascoviruses show wide levels of variation across the different host larval species. There were also significant differences in the pathogenesis of the host larvae species infected with different ascovirus isolates, even though the isolates belonged to HvAV, and they shared >90% genomic similarity (18). One possible reason why these phenomena have not been widely reported for other viruses is that the host ranges of other viruses are not as wide as that of the ascoviruses. The origin of these differences in ascovirus pathogenicity was the focus of this study, as we have aimed to determine if they are directly related to the structural differences in the virions.

The transmission of ascoviruses in the field, as previously noted, predominantly depends on the oviposition behavior of parasitic wasps. This means that ascoviruses are pathogenic microorganisms with a relatively low infection dosage (18), that is, it only requires a few virus particles to accomplish the initial infection, and this results in the host larval hemolymph being filled with mature ascovirus progeny and changes the color of the hemolymph to a milky white. As only a few ascoviruses are required for the initial infection, when they are faced with a unified host selection pressure during infection, the "adaptive changes" made by the virus are relatively uniform. Thus, only one ascovirus isolate may have adapted to different host larval species during infection, and the different kinds of adaptation may result in different host responses to different larval species; we have thus hypothesized that this may be the reason for the different pathogenicities in a single ascovirus isolate in different host larval species. If the ascovirus adapted to their hosts, the changes could be detected after the 7-day infection process in the host larvae, as this allows for the replication of 20–30 generations (approximately, from thousands of virions to $10^9$–$10^{12}$ virions).

Based on the above assumptions, this study has aimed to conduct a detailed exploration of the "self-adaptation" processes in Heliothis virescens ascovirus 3h (HvAV-3h) in response to different host larval species. HvAV-3h was isolated in China and stored in the laboratory as HvAV-3h-containing hemolymph that was naturally isolated from the host, *Spodoptera exigua* larvae. Our results showed that the components of the virion structural proteins in different host larvae were significantly different, and further validation revealed that even the structural proteins found in viral particles from different host larvae had certain differences in their interaction patterns. These differences can directly or indirectly lead to significant differences in the way virions interact with host proteins. The results of this study have revealed the origin for the differences in pathogenicity of a virus isolate in different host species from the perspective of the

changes in the composition and structure of the virus particles themselves. This has provided novel insights into the adaptability and variability of viruses and the varied pathogenicity in the hosts resulting from viral adaptability or variability.

## RESULTS

### Increases in host larvae life spans with HvAV-3h infections show species variations

HvAV-3h was inoculated into larval hemocoels to compare pathogenicity in different species. The inoculation of HvAV-3h led to 100% mortality in *M. separata*, *S. exigua*, *S. frugiperda*, and *S. litura* larvae and 95.83% ± 4.81% mortality in *H. armigera* (Fig. 1A). The *H. armigera*, *M. separata*, *S. exigua*, *S. frugiperda*, and *S. litura* were found to have larval life spans that were extended by 28, 27, 16, 26, and 17 d, which were 2.15, 1.69, 1.60, 2.17, and 1.21 times and significantly longer than those of the control check (CK) larvae, respectively (Fig. 1A).

The extension of the five larval stages was varied among the different species (Table S3), and the HvAV-3h-infected *H. armigera* had the highest $ST_{50}$ value (15.39 ± 0.50 d), which was approximately twice the $ST_{50}$ value (8.53 ± 0.12 d) of the CK *H. armigera* larvae (log-rank $\chi^2$ = 148.6, *d.f.* = 1; *P* < 0.0001), while the *S. exigua* had the lowest $ST_{50}$ value (7.89 ± 0.37 d), which was less than 1 d more than the $ST_{50}$ value (6.93 ± 0.11 d) of the CK *S. exigua* larvae (log-rank $\chi^2$ = 9.722, *d.f.* = 1; *P* < 0.0018). The extended larval stages in the virus-infected host larval species also correlated with longer molting periods, especially in the fourth and fifth instars. (Table S4).

The minimum infectious dose of HvAV-3h in *H. armigera*, *M. separata*, *S. exigua*, *S. frugiperda*, and *S. litura* was $10^{-4}$, $10^{-7}$, $10^{-7}$, $10^{-5}$, and $10^{-5}$, and this resulted in larvae mortalities of 1.39% ± 2.41%, 1.39% ± 2.41%, 0.11% ± 0.19%, 4.17% ± 4.17%, and 50% ± 4.17%, respectively (Fig. 1B). The *M. separata* larvae were thus the most sensitive to the HvAV-3h infection of the species tested, and the *H. armigera* larvae were the most resistant.

### Different host larvae-produced HvAV-3h virions had similar "principal components" but different "rare components"

A large number of virions were observed in the hemocytes of the five infected larval species (Fig. 1C). These virions were allantoic and concentrated in a certain area of the hemocytes. Furthermore, five different original HvAV-3h virions were purified from the collected hemolymph by sucrose density gradient centrifugation (Fig. 1D). Purified virions were negatively stained and observed by transmission electron microscopy (TEM) (Fig. 1E). The purity was highest for those isolated from *M. separata* and lowest for those isolated from *H. armigera*. Morphologically, there were no significant differences between the virions purified from the different host larvae.

The total protein of the HvAV-3h virions purified from the hemolymphs of different host larvae was extracted to analyze the compositions of the different original virions (Fig. 2). The background of the negative staining TEM images of the virions purified from the hemolymph of HvAV-3h-infected *H. armigera* were not clear enough, which indicated that the purity was insufficient and not suitable for further mass spectrometric (MS) analysis. Consequently, only four different original virions were analyzed. The sodium dodecyl sulfate-polyacrylamide gel electrophoresis (SDS-PAGE) analysis of the protein samples showed clear bands, and there were no significant differences in the main protein bands between the three biological replicates, which were then used for subsequent MS analysis (Fig. 2A).

After removing peptides with counts of ≤3, the MS results identified about 58–60, 63–80, 42–57, and 48–66 HvAV-3h coding proteins from the three *M. separata*, *S. exigua*, *S. frugiperda*, and *S. litura* samples (Tables S5 to S16), respectively. The results suggest that the major structural proteins encoded by HvAV-3h in the virions produced by *S. exigua*, *S. frugiperda*, and *S. litura* were similar and were mainly composed of 3H-57, 3H-152, 3H-27,

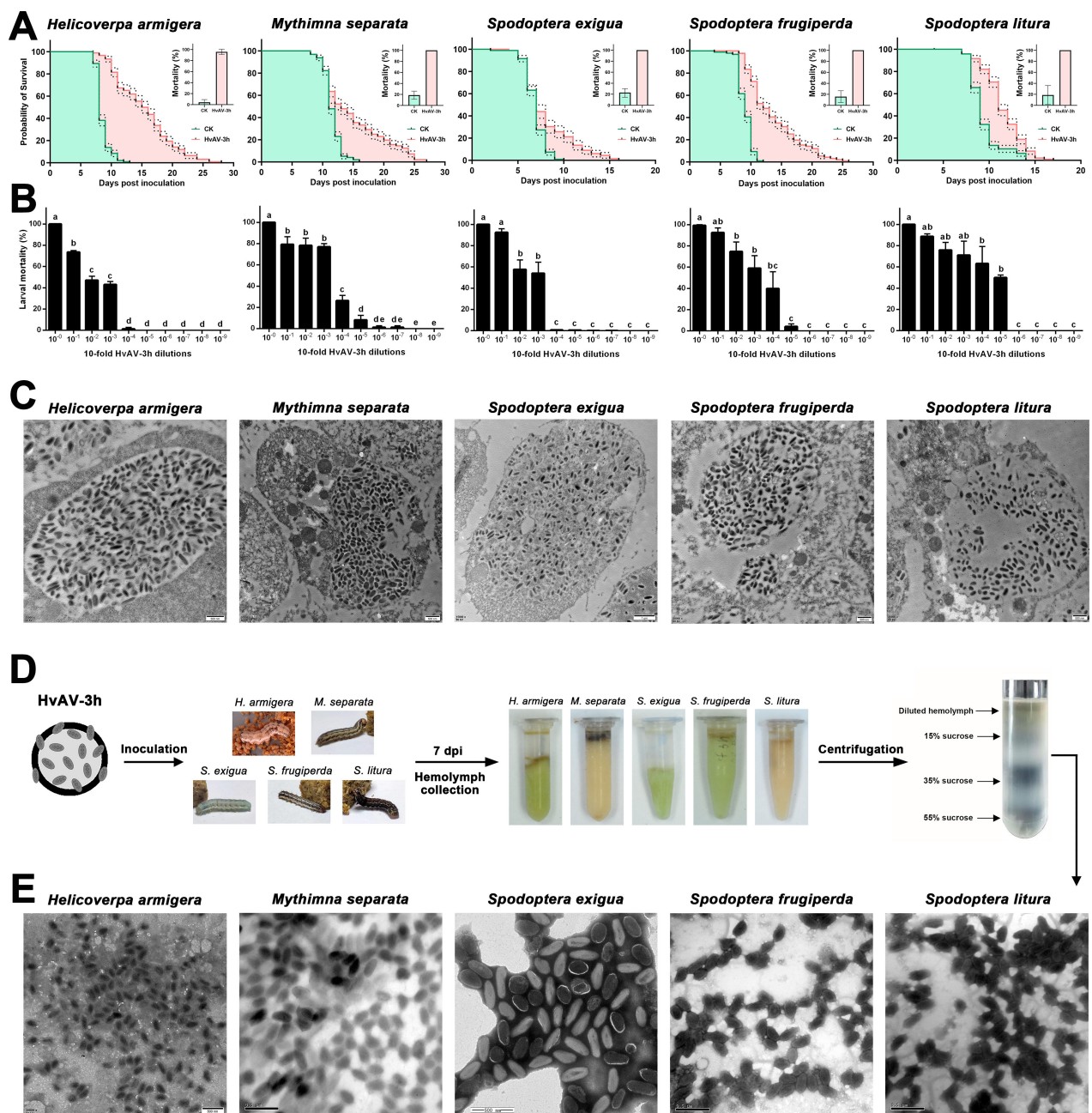

FIG 1 Comparisons of infectivity and morphology of HvAV-3h in five different host larval species. (A) Survival curves of *Helicoverpa armigera*, *Mythimna separata*, *Spodoptera exigua*, *Spodoptera frugiperda*, and *Spodoptera litura* larvae infected by HvAV-3h. The dots around the curves indicate 95% confidence intervals for each curve. The mortality of each tested larval species is provided in the upper right corner of each chart. (B) Mortalities of *H. armigera*, *M. separata*, *S. exigua*, *S. frugiperda*, and *S. litura* larvae inoculated with different diluted HvAV-3h-containing hemolymph (mean ± SE). Lowercase letters indicate significant differences between tested time points in each virus-infected larvae based on one-way analysis of variance (ANOVA) followed by least significant difference (LSD) comparisons (α = 0.05). (C) Transmission electron microscopic (TEM) observation of different host larval hemocytes after HvAV-3h infection [168 hours post infection (hpi)]. (D) Schematic diagram of the purification of HvAV-3h virions. Briefly, the HvAV-3h was first inoculated into the hemocoels of the third-instar *Helicoverpa armigera*, *Mythimna separata*, *Spodoptera exigua*, *Spodoptera frugiperda*, and *Spodoptera litura* larvae; at 7 days post infection (dpi), the hemolymph was collected separately; after ultrasonic crushing, the virus-hemolymph mixtures were loaded on the upper layer of the prepared gradient sucrose solution followed by ultracentrifugation and virion collection. (E) TEM observations showing the morphology of HvAV-3h virions purified from the hemolymph of different HvAV-3h-infected host larvae (negative staining).

3H-13, 3H-53, and 3H-58. The proportions of the first three proteins were also found to be relatively fixed (3H-57:3H-152:3H-27, approximately 2.5:1.25:1). However, the main

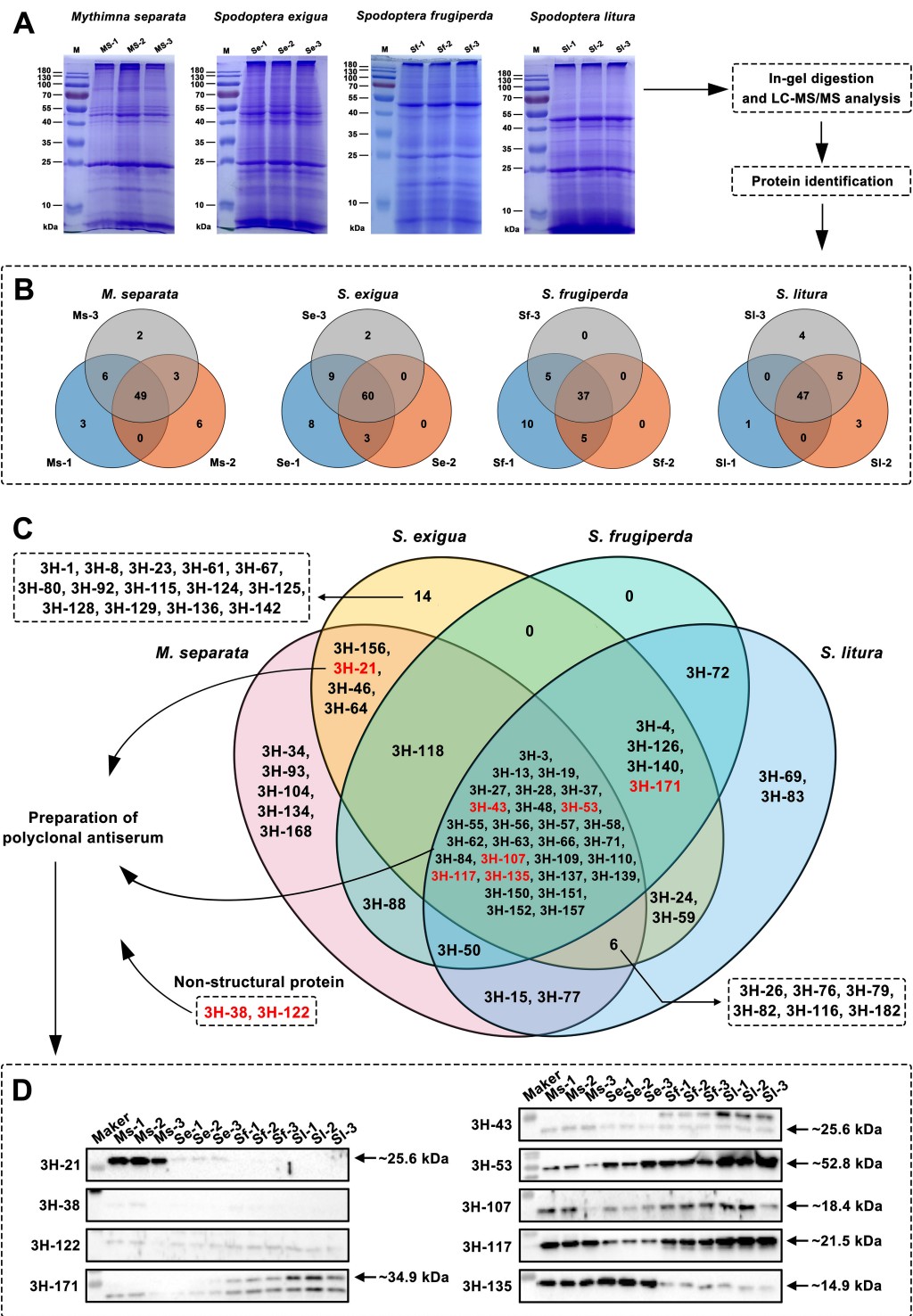

**FIG 2** Analysis of the HvAV-3h-encoded structural proteins of the virions produced by different host larval species. (A) Extraction and SDS-PAGE analysis of the total protein from different host larval species produced HvAV-3h virions. Three biological repeats were performed, and repeated samples were recorded as Latin initials of the host larvae plus Arabic numerals. (B) Analysis between three biological replicates for each host larval species produced HvAV-3h virions. (C) Venn analysis of the differences between the structural proteins of different host larval produced virions. The viral proteins marked in red are the candidates selected for the following immunoblotting confirmation. (D) Western blotting detection of the structural proteins of the HvAV-3h virions purified from different host larvae.

structural proteins of the virions produced by *M. separata* did not include 3H-152 and 3H-13 but did include 3H-56 and 3H-55. Further analyses were performed to compare the compositions of the HvAV-3h-encoded structural proteins in different host species that produced virions. The virus-encoded virion structural proteins between the three biological repeats within each host larval species were first analyzed (Fig. 2B), and the mutually contained structural proteins of the virions produced by each host larval species were used as representatives for further analysis (Fig. 2C). A total of 29 common ORFs were shared by the virions produced by the four different larval species, while 5, 14, and 2 ORFs were unique to the *M. separata*, *S. exigua*, and *S. litura* larvae producing virions, respectively. Of the 29 shared proteins, 16 were annotated and classified based on their correlative functions, including DNA metabolism, virion assembly, signaling, sugar and lipid metabolism, cellular homoeostasis, and cell lysis. Two of these proteins, MCP (3H-53), a major virion DNA-binding protein, and sulfhydryl oxidase Erv1-like protein (3H-71), were found to be involved in virion assembly.

After the removal of peptides ≤3 spectrum, 28, 26, and 27 host proteins were detected from the three protein samples of HvAV-3h virions purified from the infected *M. separata* (detailed data were provided in Tables S17 to S19); 92, 69, and 83 host proteins were detected from the three protein samples of HvAV-3h virions purified from the infected *S. exigua* (detailed data were provided in Tables S20 to S22); 292, 192, and 173 host proteins were detected from the three protein samples of HvAV-3h virions purified from the infected *S. frugiperda* (detailed data were provided in Tables S23 to S25); 40, 24, and 27 host proteins were detected from the three protein samples of HvAV-3h virions purified from the infected *S. exigua* (detailed data were provided in Tables S26 to S28). The *M. separata*-coded proteins associated with the HvAV-3h virion were mainly identified as heat shock proteins (HSP 83, HSP 70, HSP 60, and HSP 27.2), prophenoloxidase (PPO-1 and PPO-2), V-type proton pump, catalase, epoxide hydrolase, carboxylesterase (CarE), and cytochrome P450 (P450); the *S. exigua*-coded proteins were mainly identified as HSP and PPO, as well as a variety of proteins related to ATP synthesis, metabolism and ATP proton pump, UDP glycosyltransferase, and P450; the *S. frugiperda*-coded proteins were more complex than those encoded by the other three host larvae, as they contained a large number of unknown functional proteins, different types of HSP, various proteins related to ATP synthesis, metabolism and ATP proton pump, various transport proteins, and P450; and the *S. litura*-coded proteins mainly include HSP70, HSP83, catalase, ATP synthetase, P450, and various types of glutathione S-transferase (GST).

## Confirmation of HvAV-3h-coded virion structural proteins via Western blotting and immunoelectron microscopic observations

The expression and purification of the HvAV-3h-coded structural proteins used to prepare the antiserum are shown in Fig. S1. The antisera of seven HvAV-3h-encoded virion structural proteins (3H-21, 3H-43, 3H-53, 3H-107, 3H-117, 3H-135, and 3H-171) and two HvAV-3h-encoded non-structural proteins (3H-38 and 3H-122) were used to verify the results of the Venn analysis (the selected structural proteins are marked in red in Fig. 2C). The total protein samples for the virions purified from each host larva were extracted, and the distributions of the above nine proteins in the virions produced by the different host larvae were detected using Western blotting (Fig. 2D). The Western blotting results are consistent with the Venn analysis, indicating that there are significant differences in the structural proteins of the different host larval species that produce HvAV-3h virions.

The infected hemocytes appeared to react with the 3H-21, 3H-43, 3H-53, 3H-107, 3H-117, or 3H-135 antiserum, as indicated by the position of the gold label (Fig. S2). The distribution of the gold particles in each TEM image was more concentrated around the HvAV-3h virions, and no gold labels were detected in the ultrathin hemocyte sections prepared from healthy *M. separata*. The results revealed that 3H-21, 3H-43, 3H-53, 3H-107, 3H-117, and 3H-135 are the structural proteins of *M. separata*-producing

HvAV-3h virions, which indicates that the above analysis of the structural proteins from the virions (from immunoblotting assays to MS and Venn analyses) was high reliability.

## Effects of HvAV-3h-encoded virion structural proteins on viral DNA replication and viral infectivity

The HvAV-3h blocked by different structural proteins' antisera was then inoculated into the hemolymph of the third-instar *H. armigera*, *M. separata*, *S. exigua*, *S. frugiperda*, and *S. litura* larvae, respectively (Fig. 3A). The viral DNA copies of HvAV-3h blocked by different structural proteins' antisera in different host larvae were shown in Fig. 3B, and a heat map of the viral DNA copy changes between the antiserum-blocked HvAV-3h and immunoglobulin G (IgG)-incubated HvAV-3h (used as a control) was established for a clearer comparison (Fig. 3C). There was almost no unified pattern in the roles of different structural proteins in the same host and the same structural protein in different hosts according the heat map.

Further survival curves of different virions structurally blocking HvAV-3h- or IgG-incubated HvAV-3h-infected third-instar larvae were established and shown in Fig. 4. HvAV-3h blocked by different structural proteins' antisera would lead to changes in the survival curve of the host larvae after their infection. Significant differences were found between the curves of 3H-53- or 3H-107-blocked HvAV-3h-infected *S. frugiperda* and

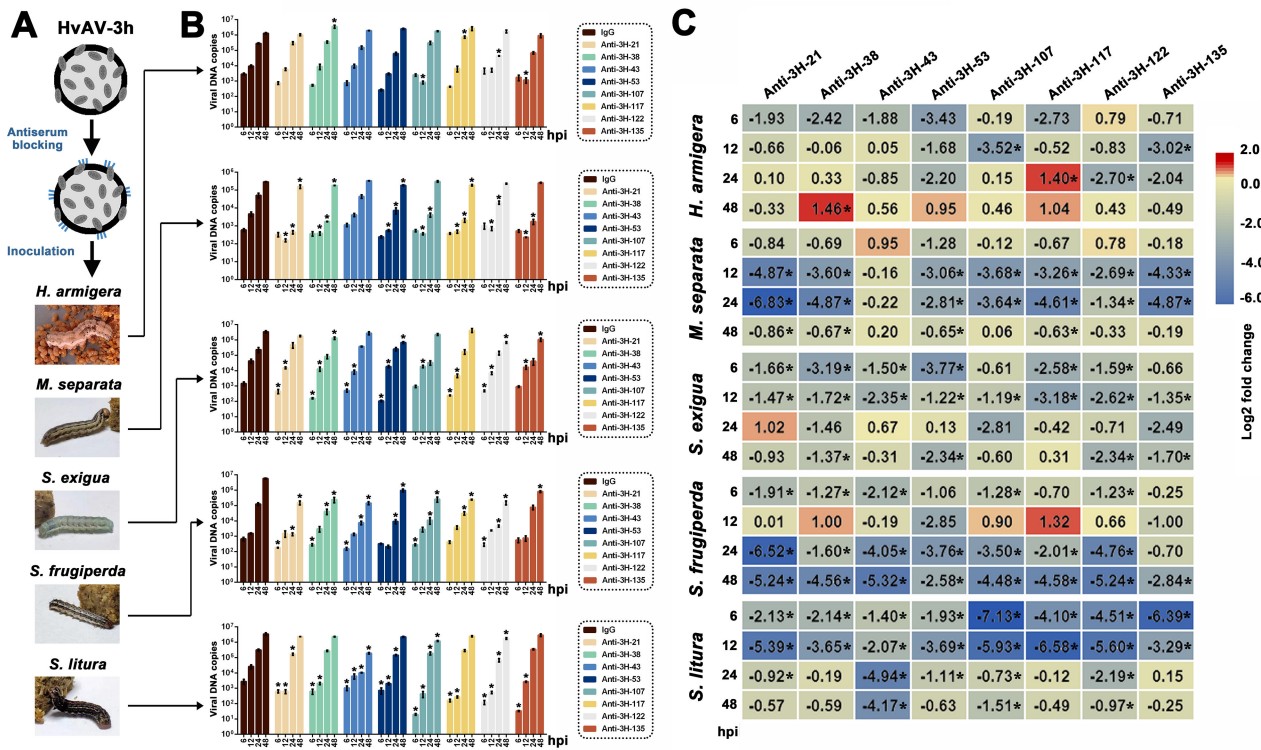

FIG 3 Effects of the HvAV-3h-coded virion structural proteins on genomic DNA replication. (A) Diagrammatic sketch of virus blocking and inoculation. The insect images are reused from Fig. 1D. The HvAV-3h was incubated with specific prepared polyclonal antibodies against 3H-21, 3H-38, 3H-43, 3H-53, 3H-107, 3H-117, 3H-122, and 3H-135, respectively. The HvAV-3h incubated with rabbit immunoglobulin G (IgG) was used as the control virus. Antibodies or IgG incubated HvAV-3h were then inoculated in the hemocoels of the third-instar *H. armigera*, *M. separata*, *S. exigua*, *S. frugiperda*, and *S. litura* larvae. (B) Absolute quantitative PCR on the HvAV-3h genomic DNA copies in different larval species (mean ± SE). Asterisks indicate the significant differences between the viral DNA copies of specific polyclonal antibody-blocked HvAV-3h and the viral DNA copies of IgG-incubated HvAV-3h at each tested time of point based on one-way ANOVA, followed by LSD comparisons (α = 0.05). (C) Heat map showing viral DNA copy fold changes after the specific HvAV-3h-coded virion structural protein was blocked. Log2 fold values between the viral DNA copies of antibody-blocked HvAV-3h and the viral DNA copies of IgG-incubated HvAV-3h at each time point were calculated and used to generate the heat map. Asterisks indicate the significant differences between the viral DNA copies of specific polyclonal antibody-blocked HvAV-3h and the viral DNA copies of IgG-incubated HvAV-3h at each time point, based on one-way ANOVA followed by LSD comparisons (α = 0.05).

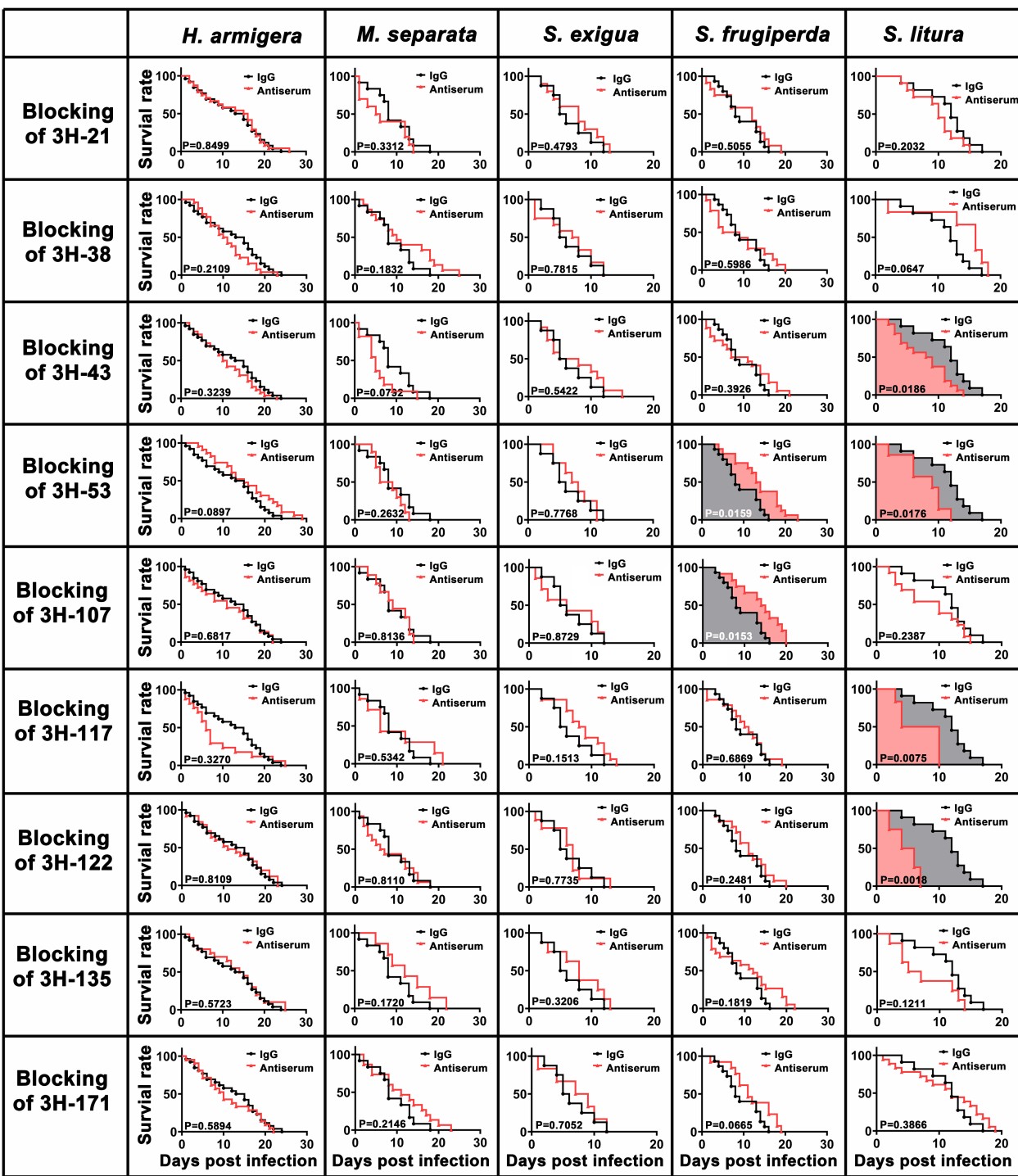

**FIG 4** Survival curves of *H. armigera*, *M. separata*, *S. exigua*, *S. frugiperda*, and *S. litura* larvae infected by specific polyclonal antibody-blocked HvAV-3h (black curves) or IgG-incubated HvAV-3h (red curves). Differences between the two curves within each tested larval species were compared using the log-rank test with SPSS 22.0, and the *P* values are provided. The survival curves with filled color indicate that significant differences were found based on log-rank test.

that of IgG-incubated HvAV-3h-infected *S. frugiperda* (all $P < 0.05$), which indicated that 3H-53- or 3H-107-blocked HvAV-3h-infected *S. frugiperda* had a higher survival rate than that of the IgG-blocked HvAV-3h-infected larvae across most of the infection period. In contrast, significant differences were found between the curves of 3H-43-, 3H-53-, 3H-117-, and 3H-122-blocked HvAV-3h-infected *S. litura* and that of the IgG-incubated

HvAV-3h-infected *S. litura* (all $P < 0.05$), which indicated that 3H-43-, 3H-53-, 3H-117-, and 3H-122-blocked HvAV-3h-infected *S. litura* had a higher survival rate than that of the IgG-blocked HvAV-3h-infected larvae across most of the infection period.

## Relationship between HvAV-3h infection and host larval detoxification enzyme activity

The previous results indicate that the virions originating from *S. exigua* and *S. frugiperda* only carried P450, virions originating from *M. separata* carried P450 and CarE, whereas those originating from *S. litura* carried both P450 and GST. These results indicate that the activity of host larval detoxification enzymes may be associated with HvAV-3h pathogenesis in different host larvae, and this further supports the relationship between the infection of HvAV-3h and the detoxification enzyme activity of the host larvae (Fig. 5). Complex patterns were found from the changes in activity of the three detoxifying enzymes in the five host larval species after infection with HvAV-3h (Fig. 5B, detailed data of larval detoxification enzyme activity are provided in Fig. S3 to S5), but two-way

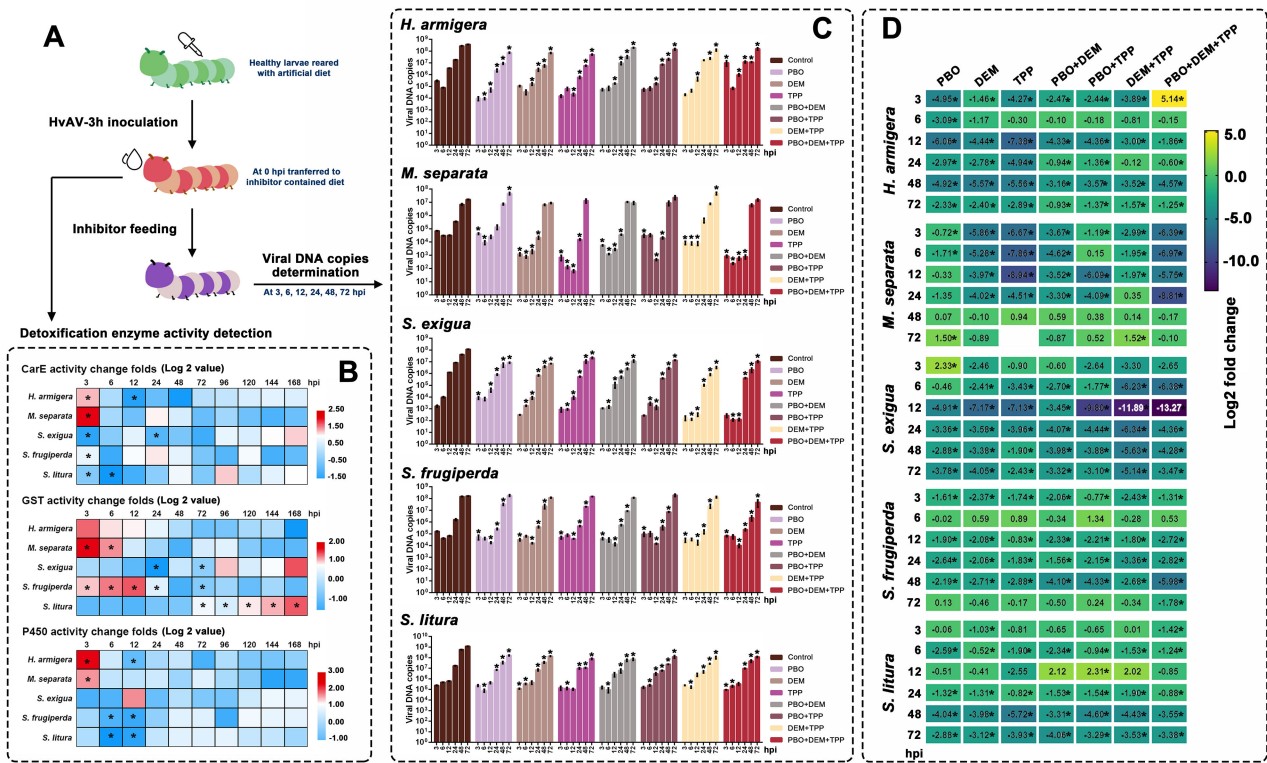

**FIG 5** Relationship of the infection of HvAV-3h and host larval detoxification enzyme. (A) Diagrammatic sketch of the treatments and larval sample collection. The third-instar *H. armigera*, *M. separata*, *S. exigua*, *S. frugiperda*, and *S. litura* larvae were inoculated with HvAV-3h, and at the specified time points post infection, the larvae were collected to perform detoxification enzyme activity determination. (B) Heat map showing the changes in larval detoxification enzyme activity after the infection of HvAV-3h. The log2 fold values between the larval detoxification enzyme activity and healthy larval detoxification enzyme activity at each tested time of point were calculated to establish the heat maps. The asterisks indicate the significant differences between the larval detoxification enzyme activity and healthy larval detoxification enzyme activity at each tested time of point based on one-way ANOVA followed by LSD comparisons ($α = 0.05$). (C) Absolute quantitative PCR on the HvAV-3h genomic DNA copies in different larval species reared with detoxification enzyme activity inhibitor supplied-artificial diets (mean ± SE). Asterisks indicate significant differences between the viral DNA copies of HvAV-3h in the larvae fed with inhibitor-supplied diet and the viral DNA copies of HvAV-3h in the larvae fed with inhibitor-free diet at each time point based on one-way ANOVA followed by LSD comparisons ($α = 0.05$). (D) Heat map of viral DNA copy fold changes of HvAV-3h inoculated in the larvae fed with inhibitor-supplied diets. Log2 fold values between the viral DNA copies of HvAV-3h in the larvae fed with inhibitor-supplied diet and the viral DNA copies of HvAV-3h in the larvae fed with inhibitor-free diet at each tested time of point were calculated, followed by heat map establishment. Asterisks indicate significant differences between the viral DNA copies of HvAV-3h in the larvae fed with inhibitor-supplied diet and the viral DNA copies of HvAV-3h in the larvae fed with inhibitor-free diet at each time point based on one-way ANOVA, followed by LSD comparisons ($α = 0.05$).

analysis of variance (ANOVA) suggested that the HvAV-3h infection resulted in the up-regulation of CarE activity in *M. separata* ($P = 0.0013$) and the down-regulation of CarE activity in *S. exigua* and *S. litura* (all $P < 0.0001$); the GST activity of *M. separata*, *S. frugiperda*, and *S. litura* was up-regulated after HvAV-3h infection (all $P < 0.0001$), as was the P450 activity in *M. separata* ($P < 0.0001$), while the P450 activity in *S. frugiperda* and *S. litura* was down-regulated (all $P < 0.001$).

The piperonyl butoxide (PBO), diethylmethoxyborane (DEM), and triphenyl phosphate (TPP) were used as inhibitors of host larval P450, GST, and CarE, respectively, and their effects on HvAV-3h genomic DNA replication were investigated (Fig. 5C). For a more intuitive comparison, a converted heat map was generated (Fig. 5D). In most cases, HvAV-3h in the single inhibitor-fed larvae had depressed replication capacity, which indicated that the viral DNA replication of HvAV-3h required host detoxification enzyme activity. Furthermore, the inhibition effect of the combination of two or three inhibitors of the DNA replication of HvAV-3h in test larvae did not have obvious synergistic effects. In contrast, among the tested *H. armigera*, *M. separata*, *S. frugiperda*, and *S. litura*, the combination of multiple inhibitors, in most cases, had the effect of mutual cancellation of the replication of the viral genome DNA. A more obvious synergistic effect of the combined application of inhibitors was found in *S. exigua* at 12 hours post infection (hpi), and the combined application of DEM + TPP and PBO + DEM + TPP made the copy number of the viral genomic DNA notably lower than that in the control larvae (reduced by $10^3$–$10^4$ times).

## Relationship between HvAV-3h structural proteins and host larval detoxification enzyme activity

The results show that the replication of ascoviruses in the host larvae is affected by the host detoxification enzyme activity, and some detoxification enzyme proteins may also be carried by ascovirus virions; thus, a series of experiments were performed to analyze the relationship between HvAV-3h structural protein and host larval detoxification enzyme activity. Four of the structural proteins (3H-43, 3H-53, 3H-107, and 3H-117) that had a significant impact on the replication of viral DNA and the survival time of host larvae were blocked and then used to infect host larvae, so as to determine their effects on the host larval detoxification enzyme activities (Fig. 6A). A heat map shows that the changes in CarE activity patterns were similar to the changes in P450 activity, whereas the change of the two enzymes was quite different from that of GST activity resulting from the blocking of the four structural proteins. These phenomena were clearly reflected in *H. armigera*, *M. separata*, *S. frugiperda*, and *S. litura*; however, the changes in the patterns of CarE and P450 for the tested *S. exigua* were not as similar as those of the other four tested larval species and were obviously different when compared with the GST. Detailed data on the detoxification enzyme activities of larvae infected with antiserum-blocked HvAV-3h are provided in Fig. S6 to S10.

To further investigate the possible relationship between the viral DNA replication of HvAV-3h and host detoxification enzyme activity, a correlation matrix analysis was performed to assess the changes in the viral DNA copies caused by infection with HvAV-3h blocked by different structural proteins' antisera (3H-43, 3H-53, 3H-107, and 3H-117) (Fig. 3C) and the subsequent changes in host larval detoxification enzyme activities (Fig. 6A). The correlation matrix analysis of *H. armigera*, *M. separata*, and *S. frugiperda* assessed the changes in the viral DNA copy number and host CarE activity and found that they were similar changes in viral DNA copy number and host P450 activity (Fig. 6B). For *H. armigera* and *S. frugiperda*, the changes in the viral DNA copy number showed mostly positive correlations with the changes in host CarE and P450 activity and mostly negative correlations with the changes in host GST activity; in *M. separata*, the changes in viral DNA copy number showed mostly negative correlations with the changes in host CarE or P450 activity and mostly positive correlations with the changes in host GST activity. In *S. litura*, the viral DNA copy number was mostly positively correlated with the changes in host CarE or P450 activity and mostly negatively

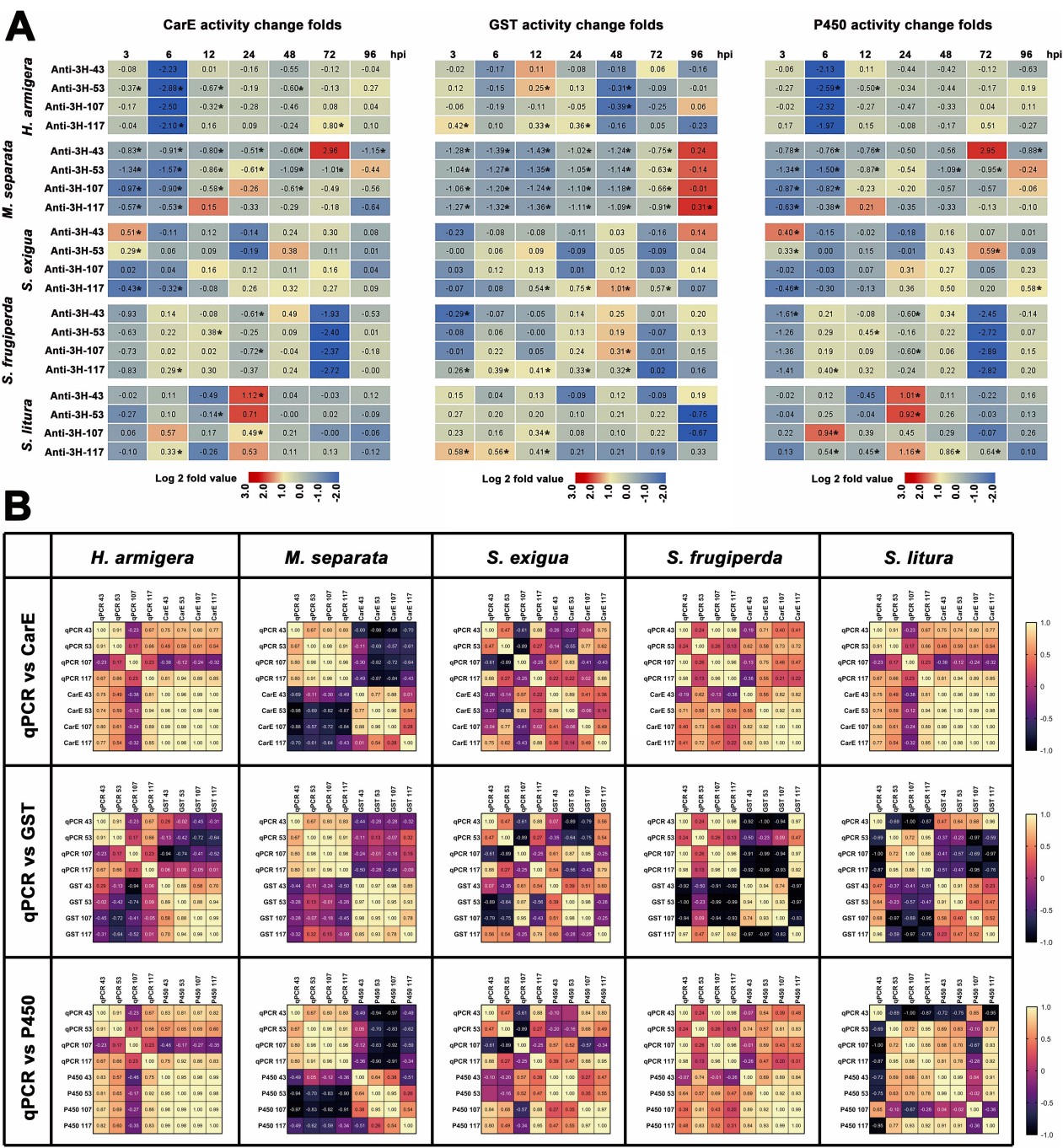

**FIG 6** Effects of HvAV-3h-coded virion structural protein on host larval detoxification enzyme activity. (A) Heat map showing the changes in detoxification enzyme activity between the larvae infection by specific polyclonal antibody (3H-43, 3H-53, 3H-107, or 3H-117)-blocked HvAV-3h and the larvae infected by IgG-incubated HvAV-3h. Log2 fold values between the detoxification enzyme activity of larvae infection by specific polyclonal antibody-blocked HvAV-3h and the larvae infected by IgG-incubated HvAV-3h at each time point were calculated and used to generate the heat maps. Asterisks indicate significant differences between the detoxification enzyme activity larvae infection by specific polyclonal antibody-blocked HvAV-3h and the larvae infected by IgG-incubated HvAV-3h at each time point based on one-way ANOVA followed by LSD comparisons ($\alpha = 0.05$). (B) Correlation matrix analysis between the viral virions structural proteins and host larval detoxification enzyme activity. The changes in the viral DNA copies caused by the virion structural protein blocking (Fig. 6C) and changes in host larval detoxification enzyme activity due to the infection of the virion structural protein blocking (Fig. 6A) were used to perform the correlation matrix analysis.

correlated with the host changes in GST activity. In *S. exigua*, the changes in viral DNA copy number and the changes in the three host detoxification enzyme activities were the weakest among the five tested larval species, which was similar to the disorder in the

activity changes of the three detoxification enzymes of *S. exigua* caused by infection with the structural protein-blocked HvAV-3h.

## Different viral structural protein interactions were found in the virions produced in different larval species

Obvious differences in the functions of the four tested structural proteins were found in the virions produced by different host larvae, according to the above correlation matrix analysis (Fig. 6B). Coimmunoprecipitation assays were performed to investigate possible interactions between these four structural proteins in different host larvae. Specific protein bands consistent with the previously obtained results (Fig. 2D) were detected for all lysates (Fig. 7). The heavy (approximately 60 kDa) and light chains (approximately 20 kDa) of rabbit IgG were detected in the immunoprecipitated lanes because of the direct binding of the secondary antibodies to rabbit IgG. Rabbit IgG did not precipitate any specific protein, indicating that the precipitate protein bands were attributed to interactions with bait proteins. In *M. separata*, larval hemolymph samples were coimmunoprecipitated with anti-3H-43-precipitated 3H-117, anti-3H-53-precipitated 3H-117, anti-3H-107-precipitated 3H-117, and anti-3H-117-precipitated 3H-43 or 3H-53. In *H. armigera*, larval hemolymph samples were coimmunoprecipitated with anti-3H-43-precipitated 3H-107 and 3H-117; anti-3H-53-precipitated 3H-107 and 3H-117; anti-3H-107-precipitated 3H-43, 3H-53, and 3H-117; and anti-3H-117-precipitated 3H-43, 3H-53, and 3H-107. In *S. exigua*, larval hemolymph samples were coimmunoprecipitated with anti-3H-107-precipitated 3H-117 and anti-3H-117-precipitated 3H-53. In *S. litura*, larval hemolymph samples were coimmunoprecipitated with anti-3H-43-precipitated 3H-53 and anti-3H-107-precipitated 3H-53. However, no bands were detected during coimmunoprecipitation assays in *S. frugiperda* larval hemolymph samples between the four tested structural proteins.

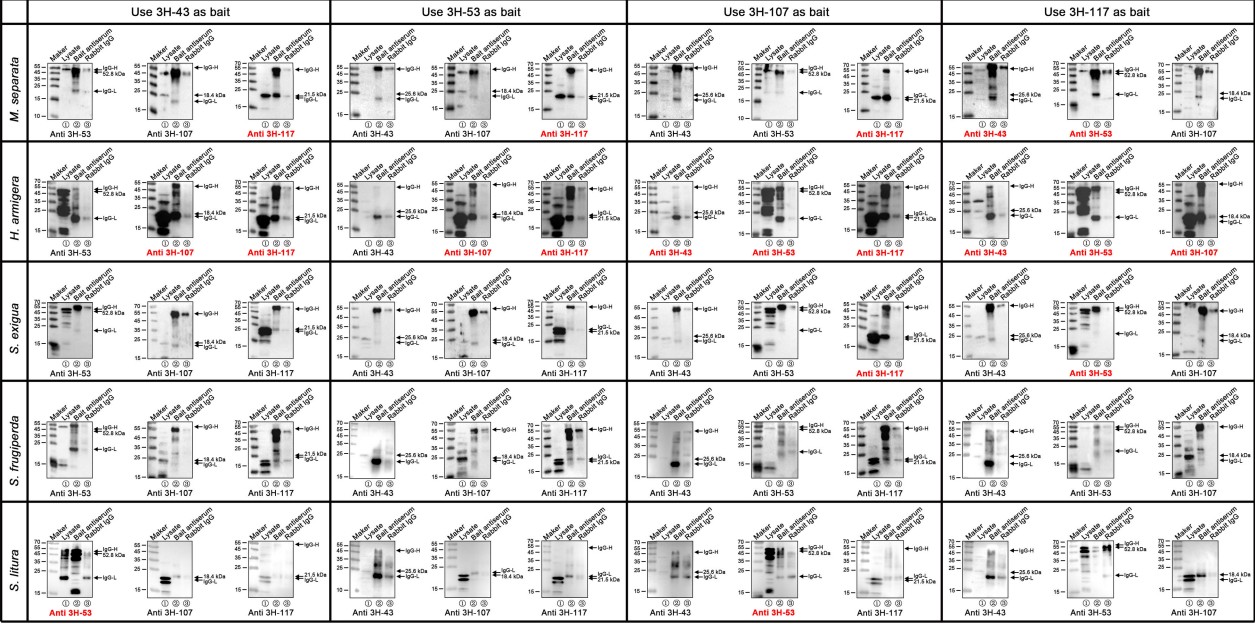

**FIG 7** Coimmunoprecipitation assays for 3H-43, 3H-53, 3H-107, and 3H-117 in different HvAV-3h-infected larval species. Lysates of the treated larval hemolymph were immunoprecipitated (IP) with antiserum against 3H-43, 3H-53, 3H-107, or 3H-117 or with rabbit immunoglobulin (IgG; used as a negative control). Samples were separated by SDS-PAGE, blotted, and probed with the indicated antibodies. Lane M, PageRuler Prestained Protein Ladder; lane ①, the lysates of the treated larval hemolymph; lane ②, the flow through of IP with specific antiserum; lane ③, the flow through of IP with rabbit IgG.

## Host GST proteins interacted with different viral proteins in the larval species

The full-length host GST proteins (Sf-GST and Sl-GST) with a 6× His tag at their N-terminus were expressed and purified from *E. coli* (Fig. S11), and their interactions with the HvAV-3h coded proteins were further characterized. Specific protein bands consistent with previously obtained results (Fig. 2D) were detected for all lysates (Fig. 8A). An approximate 52.8-kDa immunoblotting band was found from the elution of the Sl-GST lane with detection of 3H-53 antiserum, and an approximate 21.5-kDa immunoblotting band was found from the elution of the Sl-GST lane with detection of the 3H-117 antiserum, but nothing was detected in the elution of the Sf-GST lane and the Ni-NTA pull-down products, which indicates that Sl-GST interacted with HvAV-3h-coded 3H-MCP and 3H-117, but Sf-GST did not. An approximate 18.4-kDa immunoblotting band was found from the elution of the Sf-GST lane and the elution of the Sl-GST lane with the detection of the 3H-107 antiserum, but there were similar bands in the Ni-NTA pull-down products with the HvAV-3h-infected *S. frugiperda* hemolymph lysate, which indicated that the Sl-GST interacted with the HvAV-3h-coded 3H-107, but Sf-GST did not. An approximate 40.4-kDa immunoblotting band was found from the elution of the Sf-GST lane and the elution of the Sl-GST lane with the detection of the 3H-38 antiserum, and nothing was detected in the Ni-NTA pull-down products, which indicated that both the Sf-GST and Sl-GST interacted with HvAV-3h (3H-38).

## DISCUSSION

A detailed analysis of the toxicity and pathogenicity of HvAV-3h in the larvae of five noctuid pests was conducted. The results show that infection with HvAV-3h can significantly prolong the larval stage of the host (Fig. 1A; Table S3), which is in accordance with previous studies (19–23). Ascovirus isolates can infect several host larval species (15, 17, 24) and thus do not have as restrictive a range as other families such as baculovirus isolates (11–14). This has led to the hypothesis that ascoviruses are a good of material that can be used to study in detail the different pathogenicities of a single virus to different host species. Some pathogenic microorganisms with a broad host range can cause zoonosis, such as rabies, avian influenza, foot and mouth disease, anthrax, tuberculosis, and brucellosis (25–28), and they have consequently become the focus of research relating to clinical medical treatments and drug development. For example, there have been reports indicating that the rabies virus infection process in different host cells or the sensitivity of different cell lines to the virus may vary (29–31). However, because of the relatively simple structure of RNA viruses and lack of significant changes in viral morphology observed through electron microscopy in different tissues, tissue cultures, and cell lines (32–34), few studies have analyzed structural changes in the same virus in different cell lines or host cell lines. Ascoviruses are DNA viruses that differ from RNA viruses in their relatively simple structures. In previous studies, ascovirus virions have been found to contain 7–67 structural proteins (35–37). In this study, 29 core structural proteins of the HvAV-3h virion were identified by mass spectrometry analysis (Fig. 2C), and these 29 proteins were identified in all four host larvae-produced virions, as well as other proteins that were not previously identified in the virions isolated and purified from the four hosts (flexible structural protein). The results suggest that the composition of HvAV-3h virions varied in different host larvae, although few morphological changes were observed by electron microscopy (Fig. 1C and E).

After the confirmation of the virion itself had different compositional changes in different hosts, several structural proteins of HvAV-3h were selected (including core and flexible structural proteins) to block with specific antiserum to further investigate their effects on viral DNA replication and the pathogenesis of HvAV-3h in different host species (Fig. 3). Surprisingly, among the eight selected proteins, no clear pattern was determined regarding the specific structural proteins of HvAV-3h that was applicable to all tested host larval species, whether it was the influence of a structural protein on viral DNA replication or on the pathogenesis of the host. These results suggest that more complex regulatory mechanisms beyond those involving a single protein

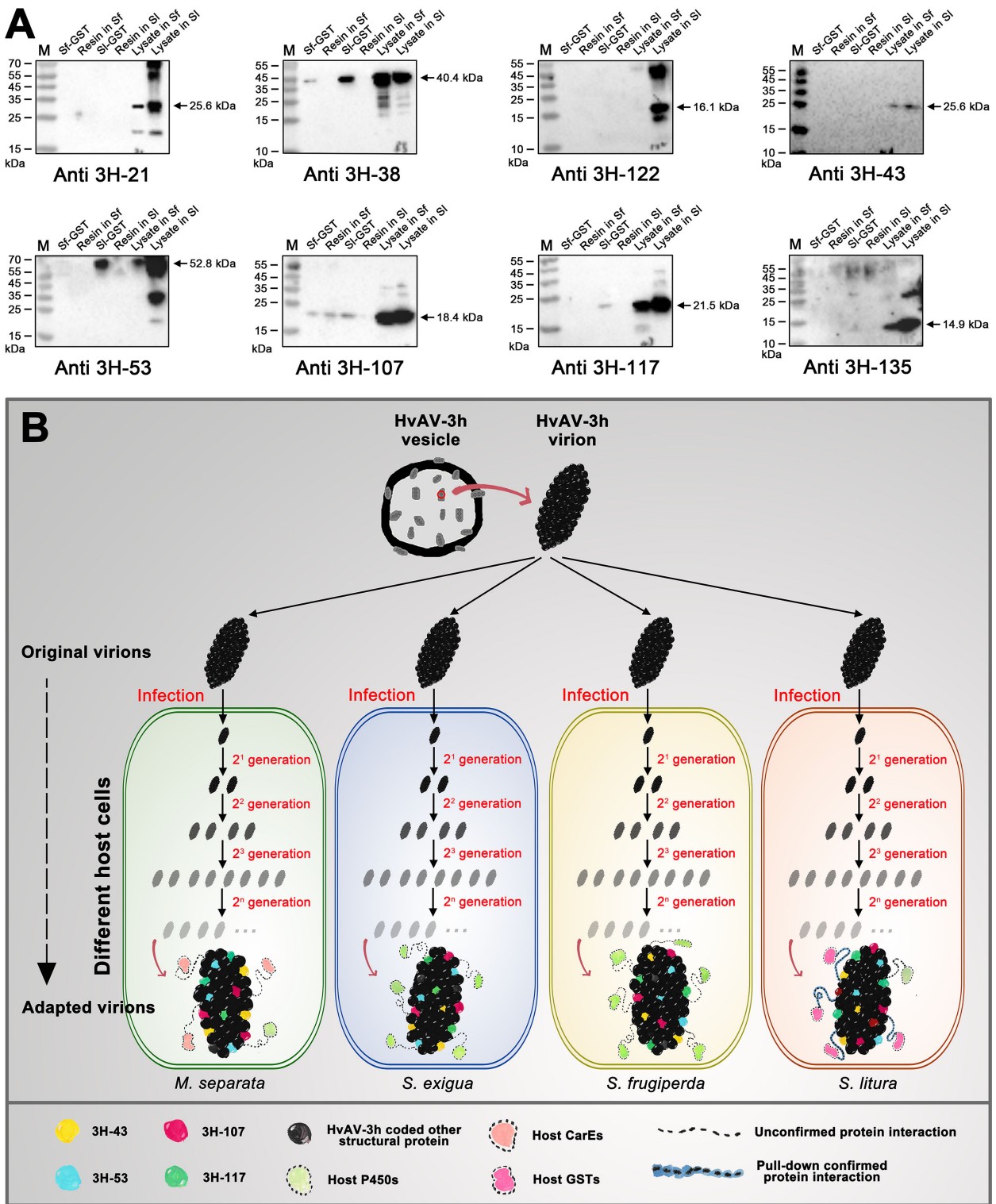

FIG 8 Confirmation of the interaction between the host GST and HvAV-3h structural proteins and a proposed model for the adapted changes of HvAV-3h in different host larval species. (A) Pull-down assay of host GST. Hemolymph of HvAV-3h-infected *S. frugiperda* or *S. litura* larvae was collected at 120 hpi to perform pull-down assays. His-Sf-GST and His-Sl-GST were incubated with the lysate from the infected *S. frugiperda* hemolymph or *S. litura* hemolymph, respectively, and the pulled-down proteins were purified with Ni-NTA resin. The Ni-NTA resin alone was used as negative control incubated with the lysate from the infected *S. frugiperda* hemolymph or *S. litura* hemolymph, respectively. The eluted samples were separated by SDS-PAGE, blotted, and probed with anti-3H-21, anti-3H-38, anti-3H-122, anti-3H-43, anti-3H-53, anti-3H-107, anti-3H-117, and anti-3H-135, respectively. (B) Proposed model for the adapted changes of HvAV-3h in different

**FIG 8** (Continued)

host larval species. The original HvAV-3h virions have similar composed structural proteins that embedded in the vesicles and contained in the collected *S. exigua* larval hemolymph. After being inoculated into the hemocoel of different host larval species, the viruses started to replicate their genomic DNA and produce their virion progeny under different environments, which means that the viruses were surviving against different host pressures, such as different activated host detoxification enzymes. After countless generations, differentiation was found in the structure of virions in different host larvae. The virions produced by different host larval species had major protein components (such as 3H-43, 3H-53, 3H-107, and 3H-117), but the interactions between these major composed proteins were different. Different optional components were also found in the virions produced by the different host larval species (such as 3H-21 and 3H-171). Furthermore, the virions' components had different interaction activities with host detoxification enzymes and other host proteins.

may exist between the structural proteins of the virus and its own replication, as well as the pathogenic process of the virus on the host. The replication of insect DNA viruses, such as baculoviruses, is mainly related to the expression of early genes (38), but these steps occur after the virus successfully enters the host cell and reaches a fixed organelle (such as the nucleus). If the differences in viral replication are caused by the influence of early viral genes on expression, then the function of the same viral protein in different hosts should reflect a certain similarity. This finding was inconsistent with our results. Therefore, we were more inclined to believe that the impact of these structural proteins on viral DNA replication was likely due to their different effects on the recognition, binding, and entry of virions during the initial stage of entry into the host cell. However, research on ascoviruses is currently limited, and there is no detailed information available regarding the steps for the recognition, binding, and entry of an ascovirus isolate. The host proteins interacting with virions, however, indicate that the differences in the internal environments of the different hosts may also explain the replication and pathogenicity differences of ascoviruses. In our previous research, we found that HvAV-3h infection could activate the melanization reaction of host larvae, and the activated melanization reaction could further promote the replication of ascoviruses (39). Prophenol oxidase (PPO) was identified in the host-associated proteins of *M. separata*, *S. exigua*, and *S. frugiperda* that produce HvAV-3h virions, indicating that PPO may directly interact with several structural proteins of the virions. Additionally, a large number of heat shock proteins, V-type proton pumps, catalase, and detoxification enzymes were identified as host-encoded HvAV-3h virion-associated proteins, which has led us to believe that these enzymes may have special significance in the process of viral infection, such as providing certain specific oxidation or reduction conditions or supplying intermediate products for the reactions.

Detailed detoxification enzyme activity analysis confirmed that ascovirus infection had a significant impact on host larval detoxification enzyme activity. Insect detoxification enzymes are believed to be directly involved in the metabolism of plant secondary metabolites and chemical pesticides in insects (40–42). Furthermore, some studies have shown that insect detoxification enzymes play important roles in the biosynthesis and degradation pathways of insect hormones (such as ecdysone and juvenile hormones) (43–47) and participate in the regulation of insect growth, development, reproduction, and other processes. After infection with an insect virus, the host detoxification enzyme system undergoes different regulatory responses that are related to different types of viruses and infected host tissues. In this study, three detoxification enzymes were identified among the structural proteins of the HvAV-3h virion. Interestingly, GST only exists in virions produced by *S. litura*, while *S. exigua* and *S. frugiperda* contain different types of P450 proteins, and virions produced by *M. separata* contain different types of CarE proteins (Tables S17 to S28). The interaction between the GST encoded by *S. litura* and the viral structural proteins encoded by HvAV-3h was confirmed using pull-down assays (Fig. 8A). These results indicate that ascoviruses do encounter different environments in their various hosts and that the virus itself may be forced to undergo structural changes in response.

The results shown in Fig. 5 indicate that the replication of HvAV-3h depends on the activity of host detoxification enzymes. Combined with the above mass spectrometry results, we inferred that this may be directly related to the virion itself carrying host

detoxification enzymes. However, the relationship between the regulatory responses of host detoxification enzymes and viral DNA replication or infection requires further investigation. To explore the specific dependency between the viral DNA replication of HvAV-3h and host detoxification enzyme activity and to further explore whether the abnormality of the viral genome DNA replication ability caused by the blocking of structural virus proteins is related to the altered host detoxification enzyme activity, we further conducted a matrix correlation analysis on the changes in HvAV-3h virus DNA replication ability and host detoxification enzyme changes after the blocking of different structural proteins (Fig. 6B). The results showed that the relationship between the three detoxifying enzymes in different host larvae and viral infection/replication was not entirely the same. However, the correlation between changes in viral DNA copy number and changes in host CarE activity or P450 activity was similar, whereas the correlation between changes in viral DNA copy number and changes in host GST activity was significantly different from the correlation between changes in the other two host detoxifying enzymes. This law was applicable to all four test insects (*H. armigera*, *M. separata*, *S. exigua*, and *S. litura*) but not applicable to the test *S. litura*, which may be because the virion produced by *S. litura* predominantly hijacked the host GST protein and the other larva-produced virions mainly carried host CarE or P450 proteins.

These results show that changes in the structural protein components of ascoviruses in different hosts or in the interaction patterns between structural protein components may be associated with changes in their ability to carry or hijack host detoxification enzymes, leading to differences in the detoxification enzyme activity of the host. We have thus confirmed the changes in the structural protein components of the virion in different hosts through mass spectrometry and further validated whether there were changes in the interaction patterns between the viral structural protein components in different hosts using coimmunoprecipitation (Fig. 7). This result also confirmed our hypothesis that there are differences in the interaction patterns between the components of the viral structural proteins in the different hosts. The interactions between the virion structural proteins and host GST were obtained and verified by pull-down assays (Fig. 8A), and the results have facilitated the construction of a schematic diagram showing the possibility of "adaptive assembly" for ascovirus virions (Fig. 8B). The "initial virion" with the same structure (in this study, this "initial virion" was produced by *S. exigua* larvae) starts the "infection-assembly-release" cycle after invading the larvae of different hosts; as the viral progeny generated cycle begins, the virus is confronted with various "obstacles" and "pressures" of the host from infection to genome DNA replication and the assembly of structural proteins; these external forces (including various host detoxification enzymes and polyphenol oxidase) make the assembly of the virion gradually change, including the arrangement or order of core structural proteins, the addition of some flexible structural proteins, or carrying different host proteins. Finally, after countless generations of "adaptive adjustment," the structure of the virion produced by different hosts shows obvious differentiation, and this process may also lead to obvious differences in host pathogenicity and virus pathogenesis.

In conclusion, HvAV-3h was used as the research object in this study to analyze the possible reasons for the virulence differentiation of a virus isolate in different host species. To understand the differences in the pathogenicity of HvAV-3h when infecting five different noctuid larval species, a detailed comparison and various verifications of the morphological and structural changes in the HvAV-3h virions were performed. The results have revealed changes in the flexible structure of the virus and provided new insights to improve our understanding of virus adaptability and the virulence differentiation caused by the adaptation process.

## MATERIALS AND METHODS

### Insect rearing and virus storage

Laboratory colonies of *Spodoptera exigua*, *Spodoptera frugiperda*, *Spodoptera litura*, and *Helicoverpa armigera* were maintained with artificial diets at 27°C ± 1°C and a 16-h light/8-h dark photoperiod (18). The commercially obtained *Mythimna separata* colony (Henan Keyun Biopesticide Co., Ltd., China) were also maintained at the same conditions as described above. The formula for the artificial diet provided to each insect species is given in Table S1.

HvAV-3h was isolated by Huang et al. (48) and stored at −20°C. For virus inoculation, hemolymph containing HvAV-3h was used according to the method described by Yu et al. (49). To preserve the virus, the hemolymph from the ascovirus-infected larvae was collected at 7 dpi. The hemolymph had approximately $1.1 \times 10^{11}$ genome copies/mL (17) and was stored at −20°C until further use.

### Virus infection and host survival time determination

HvAV-3h was inoculated into the hemocoel of the third-instar larvae of *S. exigua*, *S. litura*, *S. frugiperda*, *H. armigera*, and *M. separata,* as the description of Li et al. (17). The inoculated larvae were reared with fresh artificial diets individually in the cells (2.0 × 2.0 × 2.5 cm) of a 24-well insect culture box. Larvae treated with healthy larval hemolymph-dipped insect pins were used as controls (CK). The treated larvae were reared separately and monitored daily until all larvae died or pupated. Twenty-four larvae were used in one biological replicate, and three to five biological replicates were conducted for the HvAV-3h inoculation and control tests, respectively. Larvae that died from mechanical damage due to inoculation were corrected using mock groups. The average larval mortality of each tested larval species was calculated, and the differences between the mortality of HvAV-3h-infected larvae and that of healthy hemolymph-inoculated larvae were compared using an independent-sample *t*-test (SPSS 22.0; SPSS, Inc., Chicago, IL, USA). Survival curves for the larvae (*S. exigua*, *S. litura*, *S. frugiperda*, *H. armigera*, and *M. separata*) inoculated with HvAV-3h were established using GraphPad Prism (v6.0, GraphPad Software). Median survival times ($ST_{50}$) and the 95% confidence limits were calculated using the Kaplan-Meier method in SPSS 22.0. The ST50 values of different treatments within each larval species were compared using the log-rank test in SPSS 22.0. The duration and viability of the larval stages of each tested larva under each treatment were calculated and subjected to one-way ANOVA using SPSS 22.0, followed by the least significant difference (LSD) method for multiple comparisons.

To determine the minimal infecting dose (MID) of HvAV-3h in the *S. exigua*, *S. litura*, *S. frugiperda*, *H. armigera*, and *M. separata* larvae according to the description of Yang et al. (18), HvAV-3h was first diluted with double-distilled water (ddH$_2$O) to generate a 10-fold series of HvAV-3h dilutions ($10^0$, $10^{-1}$, $10^{-2}$, $10^{-3}$, $10^{-4}$, $10^{-5}$, $10^{-6}$, $10^{-7}$, and $10^{-8}$ series dilutions). This series of diluted HvAV-3h was inoculated into the hemocoels of different third-instar larval species, as previously described. The mortality of each HvAV-3h dilution-inoculated larvae was then calculated and analyzed using one-way ANOVA, and the differences between the different treatments were compared with LSD methods using SPSS (version 22.0; SPSS, Inc., Chicago, IL, USA).

### TEM observations and virion purification

Fresh HvAV-3h-containing hemolymph was collected from HvAV-3h-infected *S. exigua*, *S. litura*, *S. frugiperda*, *H. armigera*, and *M. separata* larvae at 7 dpi. After centrifugation (2,000× *g*) for 10 min at 4°C, the supernatant was removed, and the hemocytes were collected and fixed with 2.5% glutaraldehyde (10×) overnight at 4°C. Samples were dehydrated, embedded, sectioned, and stained according to the method described by Xu et al. (50). The ultrathin slices were then examined using a Hitachi HT7700 transmission electron microscope at an accelerating voltage of 80 kV. The HvAV-3h virions produced by different host species were purified using the hemolymph that contained

HvAV-3h, as described by Yu et al. (24). Purified virions were negatively stained and observed using TEM (HT7700, Hitachi, Japan), as described by Zhao et al. (51).

## Virion-associated proteins of HvAV-3h produced by different host larval species

Total protein was extracted from the purified HvAV-3h virions that were produced by the different host larval species (*S. exigua*, *S. litura*, *S. frugiperda*, *H. armigera*, and *M. separata*) using radio-immunoprecipitation assay (RIPA) lysis buffer (Solarbio, China), according to the manufacturer's instructions. The protein concentration of the extracted virion samples was determined using the Pierce BCA Protein Assay Kit (Thermo Fisher Scientific), according to the manufacturer's instructions. Two micrograms per milliliter of protein sample were mixed with 5× loading buffer and incubated in boiling water for 10 min. After separation using a 12% SDS-PAGE, the protein fragments were visualized using Coomassie bright blue staining. Three repeats were performed for the protein samples obtained from the virions produced by each host larval species (virion-Se-1, virion-Se-2, virion-Se-3, virion-Sf-1, virion-Sf-2, virion-Sf-3, virion-Sl-1, virion-Sl-2, virion-Sl-3, virion-Ha-1, virion-Ha-2, virion-Ha-3, virion-Ms-1, virion-Ms-2, and virion-Ms-3).

The protein gel strips were sent to the Beijing Genomics Institute (Wuhan, China) for in-gel digestion and liquid chromatography mass spectrometer/mass spectrometer detection. Briefly, the peptides were extracted from enzymatically digested proteins at different positions on the film, mass spectrometry was then used to obtain the mass spectrum (Q-EXACTIVE HF X, Thermo Fisher Scientific, San Jose, CA) of the proteins in the gel strips, and protein identification software (Mascot 2.3.02) was used to identify the proteins in the samples (52). The UniProt protein database and HvAV-3h genomic DNA were used to identify proteins.

The identified proteins were separated into viral and host larval proteins. Venn analysis was performed to assess the differences between the three biological repeats using TBtools (v0.642) (53). The identified viral and host larval proteins were analyzed, and proteins with >3 peptides were used in the Venn analysis. Furthermore, the viral proteins identified as common proteins in the three repeats of each virion purified from the different larval species were used to perform another Venn analysis using TB tools.

## Polyclonal antiserum and immunoblotting assays

To verify the structural proteins of the HvAV-3h virions, nine HvAV-3h-encoded proteins (3H-21, 3H-38, 3H-43, 3H-53, 3H-107, 3H-117, 3H-122, 3H-135, and 3H-171) were selected as representative proteins according to the obtained mass spectrum analysis data. Specific primers were designed to amplify the coding sequence (CDs) region of *3h-43*, *3h-107*, *3h-122*, *3h-135*, and *3h-171* (primer information is provided in Table S2). The genes were amplified and subcloned into pET-28a(+) vector (NoveGen, USA) according to the description of Yu et al. (24). The His-tag fused proteins (His-3H-43, His-3H-107, His-3H-122, His-3H-135, and His-3H-171) were then induced, purified, according to the description of Yu et al. (24). The obtained His-tag fused proteins were used to prepare specific polyclonal antibody the description of Yu et al. (24).

Proteins extracted from the purified HvAV-3h virions that were produced by different host larval species, as described above, were separated using 12% SDS-PAGE and transferred to a nitrocellulose membrane (Millipore, USA). After blocking with defatted milk powder [5% milk powder in Tris buffered saline with Tween (TBST) buffer], the prepared polyclonal antiserum against 3H-43, 3H-107, 3H-135, and 3H-171, as well as the previously prepared polyclonal antiserum against 3H-21 (54), 3H-38 (55), 3H-53 (56), 3H-117 (51), and 3H-122 (57), were used as primary antibodies (dilute ratio 1:3,000). Horseradish peroxidase-conjugated goat anti-mouse IgG (1:5,000) (Millipore, Billerica, MA, USA) was used as a secondary antibody. Protein bands were visualized using Clarity Western ECL Substrate (Bio-Rad).

## Immunoelectron microscopic observations

The ultrathin slices of the hemocytes collected from the HvAV-3h-infected *S. exigua* larvae (7 dpi), as described above, were exposed to 1% hydrogen peroxide (in 0.05 mol/L TBS, pH 7.4) for 5 min at 25°C. After washing five times with TBS, the slices were floated on drops of blocking buffer [1% fetal bovine serum (FBS), 1% NaN$_3$, in 0.05 mol/L TBS, pH 7.4] for 1 h at RT. After three washes with TBS, the ultrathin slices were incubated with the prepared antiserum against the 3H-21, 3H-43, 3H-53, 3H-107, 3H-117, or 3H-135 dilution (1:100 with blocking buffer) for 2 h at RT. After washing five times with TBST, the slices were floated on protein A-gold-containing blocking buffer (1:50) for 1 h at RT. The samples were then washed thoroughly with double-distilled water (five to eight times) and air dried. Grids were then stained for 20 s with drops of aqueous 1% uranyl acetate and for 20 s with 0.2% lead citrate. The samples were viewed under a Hitachi HT7700 TEM at an accelerating voltage of 80 kV.

## HvAV-3h infectious activity after structural protein antibody blocking

To detect the possible functions of virion structural proteins during HvAV-3h viral DNA replication or infection in different larval species, the antiserum against 3H-21, 3H-38, 3H-43, 3H-53, 3H-107, 3H-117, 3H-122, or 3H-135 was used to block the specific proteins exposed on the surface of HvAV-3h vesicles. Ten microliters of the collected HvAV-3h containing *S. exigua* larval hemolymph (7 dpi) were diluted in 90 μL of phosphate buffer solution (PBS, 8.0 g NaCl, 0.2 g KCl, 1.44 g Na$_2$HPO$_4$, 0.24 g KH$_2$PO$_4$ in 1-L double-distilled water), and 2 μL of the prepared antiserum was added and mixed gently, followed by incubation at 4°C overnight. Rabbit IgG (Proteintech, Wuhan, China) incubated with HvAV-3h was used as a negative control. Blocked HvAV-3h and IgG-incubated HvAV-3h were inoculated into the hemocoel of healthy third-instar larvae (*H. armigera*, *M. separata*, *S. exigua*, *S. frugiperda*, and *S. litura*). The inoculated larvae were collected 6, 12, 24, and 48 hpi. To determine the number of viral DNA copies, the total DNA of the infected larvae was extracted using the *SteadyPure* Universal Genomic DNA Extraction Kit (Accurate Biology, China). Quantitative PCR (qPCR) was then performed as described by Yu et al. (58). Five biological repeats were performed for each larval species inoculated with each blocked HvAV-3h at each time point, and three technical repeats were performed for each biological repeat. The viral DNA content in each sample was calculated, and the differences in the viral DNA copies for the samples at each tested time point between different treatments were analyzed using one-way ANOVA, followed by LSD multiple comparisons (SPSS 22.0).

Another 96 healthy third-instar larvae (*H. armigera*, *M. separata*, *S. exigua*, *S. frugiperda*, and *S. litura*) were inoculated with antiserum-incubated HvAV-3h or rabbit IgG-incubated HvAV-3h, as described above. Larval mortality was monitored daily until the larvae died or pupated. Survival curves for each tested larval population were established using Prism GraphPad 6.0, and the log-rank test was performed to analyze the differences between the survival times of the larvae inoculated with antiserum-incubated HvAV-3h and those inoculated with IgG-incubated HvAV-3h (SPSS 22.0).

## Detoxification enzyme activity of different host larval species after HvAV-3h infection

Third-instar larvae (*H. armigera*, *M. separata*, *S. exigua* *S. frugiperda*, and *S. litura*) inoculated with HvAV-3h-containing or healthy hemolymph were collected at 3, 6, 12, 24, 48, 72, 96, 120, 144, and 168 hpi. Six larvae from each time point under each treatment were used to determine the larval detoxification enzyme activity according to the procedures described by Yang et al. (18). Six individual larvae were used as biological replicates at each tested time point for each treatment. Three technical repeats were performed for each biological repeat during the determination of enzyme activity. The standard curves used in the enzyme activity assays were established using linear regression in SPSS v22.0. Differences in enzyme activity between treatments at each tested time point were analyzed using one-way ANOVA in SPSS v22.0, followed by LSD

multiple comparisons. The log2 fold value of the detoxification enzyme activity of the HvAV-3h-infected larvae versus the detoxification enzyme activity of the control larvae was calculated, and heat maps of log2 fold values for each tested larval species were established using TBtools (GitHub) (25).

## Detoxifying enzyme inhibitor effects on the infectivity of HvAV-3h

Piperonyl butoxide (Sigma, USA), diethyl maleate (Sigma, USA), and triphenyl phosphate (Macklin, China) were used as inhibitors of cytochrome P450 monooxygenase, CarE, and GST activity, respectively (59–63). Third-instar larvae (*H. armigera*, *M. separata*, *S. exigua S. frugiperda*, and *S. litura*) were inoculated with HvAV-3h and transferred onto diet dots containing 10 mg/L PBO, 5 mg/L DEM, or 10 mg/L TPP, 10 mg/L PBO and 5 mg/L DEM (PBO + DEM), 10 mg/L PBO and 10 mg/L TPP (PBO and TPP), 5 mg/L DEM and 10 mg/L TPP (DEM + TPP), or 10 mg/L PBO, 5 mg/L DEM, and 10 mg/L TPP (PBO + DEM + TPP), respectively. The HvAV-3h-inoculated larvae fed an inhibitor-free diet were used as controls. The larvae were collected at 3, 6, 12, 24, 48, and 72 h post exposure, followed by total DNA replication and qPCR to determine the viral DNA copies, as described previously. Five biological repeats were performed at each time point, and three technical repeats were performed for each biological repeat. The viral DNA content in each sample was calculated, and the differences in the viral DNA copies for the samples at each tested time point between different treatments were analyzed using one-way ANOVA, followed by LSD multiple comparisons (SPSS 22.0).

## Detoxification enzyme activity of structural protein antibody-blocked HvAV-3h-infected host larvae

Third-instar larvae (*H. armigera*, *M. separata*, *S. exigua S. frugiperda*, and *S. litura*) were inoculated with 3H-43-, 3H-53-, 3H-107-, and 3H-117-blocked HvAV-3h. Larvae inoculated with rabbit IgG-incubated HvAV-3h were used as controls. The larvae were collected at 3, 6, 12, 24, 48, 72, and 96 hpi, followed by the determination of detoxification enzyme activity, as described previously. Six individual larvae were used as biological replicates at each tested time point for each treatment. Three technical repeats were performed for each biological repeat during the determination of enzyme activity. Differences in enzyme activity between treatments at each tested time point were analyzed using one-way ANOVA in SPSS v22.0, followed by LSD multiple comparisons.

## Correlation analysis of host detoxification enzyme activity and HvAV-3h viral DNA replication

Changes in the host larval detoxification enzyme activity after infection with structural protein-blocked HvAV-3h and the changes in the viral DNA copy changes of 3H-43-, 3H-53-, 3H-107-, or 3H-117-blocked HvAV-3h in different host larval species were used to establish a correlation matrix analysis. First, the log2 fold values of host larval detoxification enzyme (GST, CarE, and P450) activity after infection with 3H-43-, 3H-53-, 3H-107-, or 3H-117-blocked HvAV-3h versus infection with IgG-incubated HvAV-3h for 6, 12, 24, or 48 hpi were calculated. Second, the log2 fold values of the viral DNA copies of 3H-43-, 3H-53-, 3H-107-, or 3H-117-blocked HvAV-3h in different larval species versus those of IgG-incubated HvAV-3h at 6, 12, 24, and 48 hpi were calculated. Finally, a correlation matrix analysis was performed to estimate the relationship between the log2 fold changes in host larval detoxification enzyme activity and viral DNA copies using GraphPad Prism 9.0.0 (121).

## Coimmunoprecipation assays for HvAV-3h-coded virion structural proteins

The hemolymph of HvAV-3h-infected *H. armigera*, *M. separata*, *S. exigua*, *S. frugiperda*, and *S. litura* was collected 120 hpi. One-milliliter Western and IP cell lysates (Beyotime, Shanghai, China) were added into 50 µL of hemolymph, and the total protein, extracted according to the manufacturer's instructions. The swollen cells were then collected and

sonicated in an ice-cold water bath and centrifuged at 12,000 × $g$ for 15 min at 4°C. The total protein concentration of the supernatant was determined by using a Pierce BCA Protein Assay Kit (Thermo Fisher Scientific) according to the manufacturer's instructions. Five microliters of the extracted protein samples (with protein content of 1.0–1.5 mg/mL) were mixed with the indicated antiserum against the bait protein (3H-43, 3H-53, 3H-107, or 3H-117) and incubated at 4°C overnight. The protein sample incubated with rabbit IgG was used as a negative control to exclude false-positive interaction bands caused by the binding to rabbit IgG. Immunoprecipitation was performed using Pierce Protein A/G magnetic beads (catalog number: 88803; Thermo Fisher Scientific). The eluates of the samples from each incubation were collected for immunoassays, as described previously. Antisera against 3H-53, 3H-107, and 3H-117 were used as primary antibodies to detect existence of the indicated protein when 3H-43 was used as bait protein; antisera against 3H-43, 3H-107, and 3H-117 were used as primary antibodies to detect existence of the indicated protein when 3H-53 was used as bait protein; antisera against 3H-43, 3H-53, and 3H-117 were used as primary antibodies to detect existence of the indicated protein when 3H-107 was used as bait protein; antisera against 3H-43, 3H-53, and 3H-107 were used as primary antibodies to detect existence of the indicated protein when 3H-117 was used as bait protein. The immunoblotting assays were performed according to the procedures as described above. The interaction was confirmed when the specific immunized band(s) as that in the lysate lane were observed in the lane of eluates of antisera against bait protein, but not found in the lane of eluates of IgG.

## Confirmation of the interaction between HvAV-3h-coded virion structural proteins and host GST using His pull-down assays

To confirm the interactions between HvAV-3h-coded virion structural proteins and host GST, the *gst* genes of *S. exigua* (*Se-gst*), *S. frugiperda* (*Sf-gst*), and *S. litura* (*Sl-gst*) were PCR amplified from the cDNA prepared from healthy or HvAV-3h-infected *S. exigua*, *S. frugiperda*, or *S. litura* larvae. Total RNA extraction and first-strand cDNA synthesis were performed according to the protocols described by Yu et al. (64). The primers used to amplify the host *gst* genes are listed in Table S1. The PCR products were purified and subcloned into the pET-28a(+) vector, as described previously. Host GST was then induced, expressed, and affinity purified according to the previously described protocol. To perform the pull-down assay, the hemolymph of HvAV-3h-infected *S. exigua*, *S. frugiperda*, or *S. litura* larvae was harvested 120 hpi, and the hemolymph of the uninfected larvae was collected and used as a control in the following analysis. Total protein was extracted from Western blots and IP cell lysates (Beyotime, Shanghai, China), according to the manufacturer's instructions. The swollen cells were then collected and sonicated in an ice-cold water bath and centrifuged at 12,000 × $g$ for 15 min at 4°C. Two hundred microliters of the prepared supernatant (with protein content of 1.0–1.5 mg/mL) were mixed with 50-µg purified host His-Se-GST, His-Sf-GST, or His-Sl-GST, respectively, and incubated at 4°C with gentle agitation for 3–6 h. Ni-NTA resin (20 mL) was added to the mixture and incubated at room temperature (approximately 25°C) for 1 h. After washing three times with washing buffer (20 mM Tris/HCl, pH 7.9, 300 mM NaCl, 30 mM imidazole, 10% glycerol), the protein was eluted from the resin using elution buffer (20 mM Tris/HCl, pH 7.9, 300 mM NaCl, 1 M imidazole, 10% glycerol). Samples were separated using SDS-PAGE, followed by Western blotting with anti-3H-21, anti-3H-38, anti-3H-43, anti-3H-53, anti-3H-107, anti-3H-117, anti-3H-122, anti-3H-135, and anti-3H-171 antisera, respectively.

### ACKNOWLEDGMENTS

We would like to thank Prof. Dun Wang, Dr. Xiao-Hua He, and Dr. Fu-Zhen Guo from College of Plant Protection, Northwest A&F University, for their assistance with the electron microscopic observation.

This work was supported by the National Natural Science Foundation of China (32070168, 32172408, 32281340018) and Agriculture Research System of China (CARS-23-C08).

H.Y., G.C., and G.-H.H. conceptualized the study. H.Y. designed the experiments. H.Y., H.C., N.L., C.-J.Y., and H.-Y.X. performed the experiments. H.Y., H.C., N.L., and C.-J.Y. analyzed the data and generated figures. H.Y. and H.C. wrote the paper. H.Y., G.C., and G.-H.H. supervised and administrated the project and performed funding acquisition. All the authors have read and agreed to the published version of this manuscript.

## AUTHOR AFFILIATIONS

[1]Hunan Provincial Key Laboratory for Biology and Control of Plant Diseases and Insect Pests, College of Plant Protection, Hunan Agricultural University, Changsha, Hunan, China
[2]Agriculture and Rural Bureau of Xinhuang Dong Autonomous County, Huaihua, Hunan, China

## AUTHOR ORCIDs

Huan Yu http://orcid.org/0000-0002-5742-4352
Gong Chen http://orcid.org/0000-0002-7199-1396
Guo-Hua Huang http://orcid.org/0000-0002-6841-0095

## FUNDING

| Funder | Grant(s) | Author(s) |
|---|---|---|
| MOST \| National Natural Science Foundation of China (NSFC) | 32070168 | Huan Yu |
| MOST \| National Natural Science Foundation of China (NSFC) | 32281340018 | Guo-Hua Huang |
| China Agricultural Research System (CARS) | CARS-23-C08 | Guo-Hua Huang |
| MOST \| National Natural Science Foundation of China (NSFC) | 32172408 | Gong Chen |

## AUTHOR CONTRIBUTIONS

Huan Yu, Conceptualization, Funding acquisition, Investigation, Project administration, Supervision, Validation, Writing – original draft, Writing – review and editing | Hong Chen, Data curation, Investigation, Methodology, Validation, Writing – original draft, Writing – review and editing | Ni Li, Investigation, Validation, Writing – review and editing | Chang-Jin Yang, Data curation, Investigation, Methodology, Writing – review and editing | Hua-Yan Xiao, Investigation, Validation, Writing – review and editing | Gong Chen, Conceptualization, Funding acquisition, Supervision, Writing – review and editing | Guo-Hua Huang, Conceptualization, Funding acquisition, Project administration, Supervision, Writing – review and editing

## DATA AVAILABILITY

Data used in this study are contained within the article and are freely available upon request from the corresponding author.

## ADDITIONAL FILES

The following material is available online.

### Supplemental Material

**Supplemental figures (Spectrum02488-23-s0001.pdf).** Fig. S1 to S11.
**Supplemental tables (Spectrum02488-23-s0002.docx).** Tables S1 to S28.

## Open Peer Review

**PEER REVIEW HISTORY (review-history.pdf).** An accounting of the reviewer comments and feedback.

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
