## [Reviewer comments · Microbiology Spectrum]

Microbiology Spectrum

Flexible changes to the *Heliothis virescens* ascovirus 3h (HvAV-3h) virion components affects pathogenicity against different host larvae species

Huan Yu, Hong Chen, Ni Li, Chang-Jin Yang, Hua-Yan Xiao, Gong Chen, and Guo-Hua Huang

Corresponding Author(s): Guo-Hua Huang, Hunan Agricultural University

Review Timeline:

Submission Date:	June 14, 2023
Editorial Decision:	July 30, 2023
Revision Received:	August 7, 2023
Accepted:	August 30, 2023

Editor: Clinton Jones

Reviewer(s): Disclosure of reviewer identity is with reference to reviewer comments included in decision letter(s). The following individuals involved in review of your submission have agreed to reveal their identity: İkbâl Agah İnce (Reviewer #1)

Transaction Report:

DOI: <https://doi.org/10.1128/spectrum.02488-23>

July 30, 2023

Dr. Guo-Hua Huang
Hunan Agricultural University
College of Plant Protection
Nongda Road 1
Changsha, Hunan 410128
China

Re: Spectrum02488-23 (**Flexible changes to the *Heliothis virescens* ascovirus 3h (HvAV-3h) virion components affects pathogenicity against different host larvae species**)

Dear Dr. Guo-Hua Huang:

Thank you for submitting your manuscript to Microbiology Spectrum. As you will see your paper is very close to acceptance. Please modify the manuscript along the lines I have recommended. As these revisions are quite minor, I expect that you should be able to turn in the revised paper in less than 30 days, if not sooner. If your manuscript was reviewed, you will find the reviewers' comments below.

When submitting the revised version of your paper, please provide (1) point-by-point responses to the issues raised by the reviewers as file type "Response to Reviewers," not in your cover letter, and (2) a PDF file that indicates the changes from the original submission (by highlighting or underlining the changes) as file type "Marked Up Manuscript - For Review Only". Please use this link to submit your revised manuscript. Detailed instructions on submitting your revised paper are below.

Link Not Available

Sincerely,

Clinton Jones

Reviewer comments:

Reviewer #1 (Comments for the Author):

Dear Authors,

The Authors submitted a detailed research on the "Flexible changes to the *Heliothis virescens* ascovirus 3h (HvAV-3h) virion components affects pathogenicity against different host larvae species." I appreciate the valuable contributions of the study in the field by exploring the virus-host interaction mysteries between different hosts in one of the lesser-known virus group Ascoviridae.

After a thorough review of the work, I have two key suggestions to improve the manuscript.: the co-IP Assay section and the presentation of proteomics data in the supplementary tables.

Co-IP Assay Elaboration and Controls:

Please provide a more detailed description of the co-IP Assay methodology. Elaborate on the experimental setup, including the specific antibodies used for immunoprecipitation and the detection of interacting proteins. Additionally, clarify the controls used in the assay to ensure the reliability of your results, especially concerning false positive detection. Describing negative controls and

how they help identify non-specific interactions will strengthen the validity of your findings.

Proteomics Data and Supplementary Tables:

In the supplementary tables presenting the proteomics data, ensure that the information is comprehensive and well-organized. Provide additional relevant details on the identified proteins, such as their functional annotations and potential roles in immune responses. If any of the detected proteins are not unique to your study but are closely related to previously reported data in related virus groups, make a clear comparison to highlight any similarities or differences. This will help readers contextualize your findings within the existing knowledge on these proteins.

Overall, these revisions will enhance the clarity and robustness of your study, enabling readers to understand better your experimental approach and the significance of your results. Addressing these suggestions will significantly improve the manuscript's quality and its impact on the scientific community.

Reviewer #2 (Comments for the Author):

Ascoviruses are a group of large, double-stranded DNA viruses that mainly infect insects of the family Noctuidae. In this manuscript submitted by Yu et al. entitled "Flexible changes to the *Heliothis virescens* ascovirus 3h (HvAV-3h) virion components affects pathogenicity against different host larvae species", the authors found that the virus (HvAV-3h) infection had significant effects on the life span of five different lepidopteran insects, although the morphology of the virions purified from the infected insects was similar. Mass spectrometry analysis revealed that the virions produced by different host insects share 29 common viral proteins but also contain some distinct viral components and host proteins, particularly of some detoxification enzymes (P450, GST). Western blotting and immuno-electronic microscopy confirmed the presence of some viral proteins (3H-13, 27, 55, 56, 57, 58, 152) on the purified virions. Blocking/neutralizing the virions with the antibodies of those viral proteins reduced the viral DNA replication and virus infectivity, and to a certain extent, affected the host detoxification enzymes activities. Overall, the manuscript provided a large body of data, however, the results were not well organized and described or interpreted. It is expectable that the components of virions purified from different host insects have subtle difference, but the functional links of those components, particularly of those detoxification enzymes, with the virus infectivity are not clear.

Major points:

1. Lines 140-147, the morphology of the purified virions has no obvious difference. This section shouldn't stand alone. It could be combined with the section "Different host larvae produced HvAV-3h virions had similar....." in lines 148-149. Also, Lines 220-233, they compared the host proteins associated with the HvAV-3h virion, this section should be combined with those in lines 148-149. Additionally, the description and interpretation of the components of virions purified from different hosts are not clear.
2. Lines 250-251, they used the inhibitors of the detoxification enzymes to evaluated the relationship of the virus infectivity and the activity of enzymes. The inhibitory effect of each inhibitor on the specific enzyme of insects are not clear.
3. Lines 262-296. Since the antibody block/neutralization substantially reduced the viral DNA replication and infectivity, it is not conceivable to assess the relationship of the viral structural proteins and host detoxification enzyme activity by using the antibody blocked/neutralized virus to infect the insect.

Minor points

1. Lines 158-160, the proteins associated with the virions purified from each insect species are varied remarkably.
2. Lines 226, 228, 231-232, the difference of "P450" and "P450s" is not clear. P450 is a large protein family, which specific member(s) of P450 were detected in those purified virions?
3. Line 231, "*S. frugiperda*" should be "*S. litura*".

Preparing Revision Guidelines

For complete guidelines on revision requirements, please see the journal Submission and Review Process requirements at

<https://journals.asm.org/journal/Spectrum/submission-review-process>. **Submissions of a paper that does not conform to Microbiology Spectrum guidelines will delay acceptance of your manuscript. "**

Please return the manuscript within 60 days; if you cannot complete the modification within this time period, please contact me. If you do not wish to modify the manuscript and prefer to submit it to another journal, please notify me of your decision immediately so that the manuscript may be formally withdrawn from consideration by Microbiology Spectrum.

Response letter for Spectrum02488-23

Dear editor Clinton Jones,

Thank you for your decision letter concerning our manuscript (ID Spectrum02488-23) entitled “Flexible changes to the *Heliothis virescens* ascovirus 3h (HvAV-3h) virion components affects pathogenicity against different host larvae species” and your time regarding for our revision. I also appreciate all the critical comments from you and reviewers. We have carefully considered the comments and revised the manuscript based on your comments and suggestions. With these improvements, we hope that the current version can meet the Journal’s standards for publication. The following is a point-by-point response to all those comments and a list of changes we have made to the manuscript.

Reviewer: 1

1. Co-IP Assay Elaboration and Controls: Please provide a more detailed description of the co-IP Assay methodology. Elaborate on the experimental setup, including the specific antibodies used for immunoprecipitation and the detection of interacting proteins. Additionally, clarify the controls used in the assay to ensure the reliability of your results, especially concerning false positive detection. Describing negative controls and how they help identify non-specific interactions will strengthen the validity of your findings.

Response: Thank you for your suggestion. We had revised the method of Co-IP according to your suggestion. Please see the revised manuscript.

2. Proteomics Data and Supplementary Tables: In the supplementary tables presenting the proteomics data, ensure that the information is comprehensive and well-organized. Provide additional relevant details on the identified proteins, such as their functional annotations and potential roles in immune responses. If any of the detected proteins are not unique to your study but are closely related to previously reported data in related virus groups, make a clear comparison to highlight any similarities or differences. This will help readers contextualize your findings within the existing knowledge on these proteins.

Response: Thank you for your suggestion. We had 24 supplementary Tables to present the proteomic data, 12 for the viral proteins and 12 for host larval proteins. All them contained annotation information. Considering we do not detect the immune responses, such as melanization or other humoral immunity or cellular immunity

pathways, we did not revise these tables. Please excuse us for we did not adding any highlights.

Reviewer: 2

Comments:

Overall, the manuscript provided a large body of data, however, the results were not well organized and described or interpreted. It is expectable that the components of virions purified from different host insects have subtle difference, but the functional links of those components, particularly of those detoxification enzymes, with the virus infectivity are not clear.

Response: Thank you for your valuable comments. The study on ascovirus is relatively backward, and there are many unknowns about their pathogenesis and the structure of virions, which brings great difficulties to our work. In this study, we mainly aim to demonstrate the mutual "adaptation" between the ascovirus and the host by verifying the variability of the structure of virions. There are indeed some aspects that cannot be explained clearly in this study, and this is also what we want to study in the future. At present, we are preparing antibodies against the host detoxification enzyme proteins associated to the virions, and we will use these antibodies to further study the relationship between virus infection and host detoxification enzyme proteins and activity.

Specific points:

1. Lines 140-147, the morphology of the purified virions has no obvious difference. This section shouldn't stand alone. It could be combined with the section "Different host larvae produced HvAV-3h virions had similar....." in lines 148-149. Also, Lines 220-233, they compared the host proteins associated with the HvAV-3h virion, this section should be combined with those in lines 148-149. Additionally, the description and interpretation of the components of virions purified from different hosts are not clear.

Response: Thank you for your suggestion. We had rearranged the sections in the RESULTS according to your suggestion. The description of the mass spectrum results about the different host produced virions were separated into two parts: the protein encoded by the virus and the protein encoded by host larvae. They had been combined into a same subsection according to your suggestion. And we had added some general description about the host larval coded proteins identified from the MS data. Hope the revised manuscript could meet your requirements.

2. Lines 250-251, they used the inhibitors of the detoxification enzymes to evaluate the relationship of the virus infectivity and the activity of enzymes. The inhibitory effect of each inhibitor on the specific enzyme of insects are not clear.

Response: Thank you. PBO, DEM, and TPP were commonly used as inhibitors to the

insect detoxification enzymes, thus we did not detect their inhibitory effects in this study. We had added several references in the M&M sections. Please see the revised manuscript.

3. Lines 262-296. Since the antibody block/neutralization substantially reduced the viral DNA replication and infectivity, it is not conceivable to assess the relationship of the viral structural proteins and host detoxification enzyme activity by using the antibody blocked/neutralized virus to infect the insect.

Response: As we had discussed in the DISCUSSION, the viral DNA replication and infectivity might be directly resulted from the reduced invading of the viruses due to the blocked structural protein. But how does the blocked structural protein in the virions affect the invading of ascovirus was unknown. From the MS data we can see that the virions might carry the host detoxification enzyme protein, which suggested that the host detoxification enzyme protein should interacted with the virion structural proteins (virus coded ones). These interactions might happen in the late stage of the infection of ascovirus (the stage of assemble of virions), but this still indicates the infection of ascovirus are associated to the host larval detoxification enzyme activity. On the other side, as shown in Fig. 5B, the host larval detoxification enzyme activity changed a lot from 3-24 hpi, and this stage was the invading stage of the ascovirus, which indicates that the invading of HvAV-3h is related to the host larval detoxification enzyme activity. To reveal whether the selected viral structural protein had any functions to stimulate the host larval detoxification enzyme activity, so as to influence the invading of ascovirus, we performed the experiments of Fig. 6A (Lines 262-296).

4. Lines 158-160, the proteins associated with the virions purified from each insect species are varied remarkably.

Response: Many thanks. To identify the virion components, the MS analysis of virion protein samples purified from each larval species were performed with 3 biological repeats. As you see there are differences between the 3 repeats. The Veen analysis of the 3 repeats of each insect species were performed, and to avoid the inaccuracies caused by these differences, we used those proteins commonly identified in all the 3 repeats. We hope that the conclusion obtained in this way will be more reliable.

5. Lines 226, 228, 231-232, the difference of "P450" and "P450s" is not clear. P450 is a large protein family, which specific member(s) of P450 were detected in those purified virions?

Response: Sorry for our careless. We had uniform the "P450s" into "P450", "GSTs" into "GST", please see the revised manuscript. The specific numbers of P450 were

provided in the supplementary Tables. There are too many identified P450, and it is not appropriate to list their numbers one by one. For example, 15 P450 proteins (CYP324A6, CYP4L7, CYP6AE97, CYP4S8, CYP6AE10, CYP354A14, CYP6AB61, CYP332A1, CYP6B68, CYP339A1, CYP9A21v4, CYP6AB31, CYP306A1, CYP4M15, CYP321A9) were identified from the virions purified from *S. exigua*. Furthermore, CarE and GST also are protein families, if we list the numbers of P450, we also have to separate CarE and GST proteins by their subgroups. So, we didn't add the specific member(s) of P450 in the revised manuscript. The readers can find and select the detailed information in our provided supplementary data.

6. Line 231, "*S. frugiperda*" should be "*S. litura*".

Response: Sorry for our careless. We had corrected the mistakes. Please see the revised manuscript.

We hope the revision could meet the requirement to your journal and be better readable to authors.

Sincerely,
Dr. Guo-Hua Huang (Corresponding author)
Hunan Agricultural University,
Changsha 410128, Hunan,
China

August 30, 2023

Dr. Guo-Hua Huang
Hunan Agricultural University
College of Plant Protection
Nongda Road 1
Changsha, Hunan 410128
China

Re: Spectrum02488-23R1 (**Flexible changes to the *Heliothis virescens* ascovirus 3h (HvAV-3h) virion components affects pathogenicity against different host larvae species**)

Dear Dr. Guo-Hua Huang:

Your manuscript has been accepted, and I am forwarding it to the ASM Journals Department for publication. You will be notified when your proofs are ready to be viewed.

Sincerely,

Clinton Jones
Editor, Microbiology Spectrum
